# Oceanic response to the consecutive Hurricanes Dorian and Humberto (2019) in the Sargasso Sea

Dailé Avila-Alonso[1,2], Jan M. Baetens[2], Rolando Cardenas[1], and Bernard De Baets[2]

[1]Laboratory of Planetary Science, Department of Physics, Universidad Central "Marta Abreu" de Las Villas, 54830, Santa Clara, Villa Clara, Cuba

[2] KERMIT, Department of Data Analysis and Mathematical Modelling, Faculty of Bioscience Engineering, Ghent University, 9000 Ghent, Belgium

**Correspondence:** Dailé Avila-Alonso (davila@uclv.cu)

**Abstract.** Understanding the oceanic response to tropical cyclones (TCs) is of importance for studies on climate change. Although the oceanic effects induced by individual TCs have been extensively investigated, studies on the oceanic response to the passage of consecutive TCs are rare. In this work, we assess the upper oceanic response to the passage of the Hurricanes Dorian and Humberto over the western Sargasso Sea in 2019 using satellite remote sensing and modelled data. We found that the combined effects of these slow-moving TCs led to an increased oceanic response during the third and fourth post-storm weeks of Dorian (accounting for both Dorian and Humberto effects) because of the induced mixing and upwelling at this time. Overall, anomalies of sea surface temperature, ocean heat content and mean temperature from the sea surface to a depth of 100 m were a 50, 63 and 57% smaller (more negative) in the third/fourth post-storm weeks than in the first/second post-storm weeks of Dorian (accounting only for Dorian effects), respectively. For what concerns the biological response, we found that surface chlorophyll-a (chl-a) concentration anomalies, the mean ch-a concentration in the euphotic zone and the chl-a concentration in the deep chlorophyll maximum were 16, 4 and 16% higher in the third/fourth post-storm weeks than in the first/second post-storm weeks, respectively. The sea surface cooling and increased biological response induced by these TCs were significantly higher (Mann-Whitney test $p < 0.05$) as compared to climatological records. Our climatological analysis reveals that the strongest TC-induced oceanographic variability in the western Sargasso Sea can be associated with the occurrence of consecutive TCs and long-lasting TC forcing.

**Keywords:** Chlorophyll-a concentration, sea surface temperature, tropical cyclones

## 1 Introduction

Hurricanes and typhoons (or more generally, tropical cyclones (TCs)) are among the most destructive natural phenomena on Earth leading to great social and economic losses (Welker and Faust, 2013; Lenzen et al., 2019), as well as ecological perturbations of both marine and terrestrial ecosystems (Fiedler et al., 2013; de Beurs et al., 2019; Lin et al., 2020). Given the devastating effects of TCs, the question of how they will be affected by climate change has received considerable scientific attention (Henderson-Sellers et al., 1998; Knutson et al., 2010; Walsh et al., 2016). Modelling studies project either a decrease in TC frequency accompanied by an increased frequency of the strongest storms or an increase of the number of TCs in general,

depending on the spatial resolution of the models (Knutson et al., 2010; Camargo and Wing, 2016; Walsh et al., 2016; Zhang et al., 2017; Bacmeister et al., 2018; Bhatia et al., 2018). In any case, an assessment of the impact of climate change on future TC activity needs to be based on whether or not the past and present changes in climate have had a detectable effect on TCs to date (Walsh et al., 2016).

The ocean is the main source of energy for TC intensification, hence, changes in oceanic environments considerably affect TC activity (Knutson et al., 2010; Huang et al., 2015; Sun et al., 2017; Trepanier, 2020). Sea surface temperature (SST) and ocean heat content (OHC) have risen significantly over the past several decades in regions of TC formation (Santer et al., 2006; Defforge and Merlis, 2017; Trenberth et al., 2018; Cheng et al., 2019; Zanna et al., 2019; Chih and Wu, 2020). Accordingly, TC lifetime maximum intensity significantly increased during 1981–2016 for both the Northern and Southern Hemispheres (Song et al., 2018). More specifically, both the frequency and intensity of TCs in the North Atlantic basin increased over the past few decades (Deo et al., 2011; Walsh et al., 2016), while a significant increase in TC intensification rates in the period 1982–2009 has been documented with a positive contribution from anthropogenic forcing (Bhatia et al., 2019).

The most recent Atlantic hurricane seasons have shown well-above normal activity (Trenberth et al., 2018; Bang et al., 2019). For instance, the 2017 hurricane season was extremely active with 17 named storms (1981–2010 median is 12.0), 10 hurricanes (median is 6.5) and 6 major hurricanes (median is 2.0) (Klotzbach et al., 2018). It will be remembered for the unprecedented devastation caused by the major hurricanes Harvey, Irma and Maria breaking many historical records (Todd et al., 2018; Trenberth et al., 2018; Bang et al., 2019). This high activity was associated with unusually high SST in the eastern Atlantic region, where many storms developed, together with record-breaking OHC values favouring TC intensification (Lim et al., 2018). Besides, the 2019 hurricane season was relevant because of the high number of named storms (18 storms) and the development of the long-lasting Hurricane Dorian, which broke the record for the strongest Atlantic hurricane outside the tropics (>23.5°N) (Klotzbach et al., 2019; Ezer, 2020). Two weeks after the passage of Dorian across the western Sargasso Sea, TC Humberto moved across this area. The interaction of two storms closely related in time and space provides an in situ experiment for studying oceanic response (Baranowski et al., 2014).

Over oceans, TC-induced wind forcing mixes the surface layer, deepens the mixed layer and uplifts the thermocline leading to a decreased upper ocean temperature and heat potential (Price, 1981; Shay and Elsberry, 1987; Trenberth et al., 2018). Vertical mixing and upwelling also lead to an increased abundance of surface phytoplankton due to entrainment of nutrient-rich waters from the nitracline to the ocean surface and/or entrainment of phytoplankton from the deep chlorophyll maximum (DCM) (Babin et al., 2004; Walker et al., 2005; Gierach and Subrahmanyam, 2008; Shropshire et al., 2016). The nutrient influx stimulates phytoplankton growth and can lead to phytoplankton blooms lasting several days after the TC passage in the oligotrophic oceanic waters (Babin et al., 2004; Hanshaw et al., 2008; Shropshire et al., 2016). Moreover, rainfall associated with these extreme meteorological phenomena modulates surface cooling and phytoplankton blooms since rainfall freshens the near-surface water, thus increasing stratification and influencing vertical mixing (Lin and Oey, 2016; Liu et al., 2020). From satellite imagery, an increase in phytoplankton abundance is identified as an elevated chlorophyll-a (chl-a) concentration. Distinguishing between mechanisms inducing changes in chl-a concentration is crucial to understanding the impact of storms on upper ocean oceanographic conditions. Although there have been extensive studies investigating the oceanic

response described above to the passage of individual TCs, the effects induced by consecutive TCs have been much less documented (Wu and Li, 2018; Ning et al., 2019). More specifically, extensive and long-lasting SST cooling as well as intense post-storm phytoplankton blooms after the passage of consecutive TCs have been documented in the northwestern Pacific Ocean (e.g., Wu and Li, 2018; Ning et al., 2019; Wang et al., 2020). However, to the best of our knowledge, there are no previous studies assessing the biological response to consecutive TCs in the western Sargasso Sea.

Given that the greatest ocean warming is projected to occur by the end of the century (Cheng et al., 2019), we may anticipate a further increase in TC intensity and/or frequency. The assessment of the oceanic response to TCs has been a hot topic given its importance for studies on climate change, ecological variability and environmental protection (Fu et al., 2014). More specifically, insights into the phytoplankton response to severe weather events are essential in order to ascertain the capacity of the oceans to absorb carbon dioxide through photosynthesis (Davis and Yan, 2004). Hence, several studies have assessed the oceanic response to recent major hurricanes in the North Atlantic basin (e.g., Trenberth et al., 2018; Miller et al., 2019b; Avila-Alonso et al., 2020; Hernández et al., 2020) and others in very active hurricane seasons such as in 2005 (e.g., Oey et al., 2006, 2007; Shi and Wang, 2007; Gierach and Subrahmanyam, 2008). In this work, we assess the upper oceanic responses induced by Hurricanes Dorian and Humberto in the western Sargasso Sea in 2019. This gives insights into the implications of a simultaneous increase of both the frequency and intensity of TCs in the North Atlantic basin.

## 2 Materials and Methods

### 2.1 Study area

The Sargasso Sea is the part of the North Atlantic Ocean (known as the North Atlantic gyre) that is bounded by the surrounding clockwise-rotating system of major currents, i.e., the North Atlantic Current on the north, the Gulf Stream on the west, the North Atlantic Equatorial Current on the south, and the Canary Current on the east (Deacon, 1942; Laffoley et al., 2011). Hence, the Sargasso Sea essentially lies between the parallels 20–35°N and the meridians 30–70°W (Augustyn et al., 2013). We considered the western Sargasso Sea as our study area (shown in Figure 1), since it was affected by Dorian and Humberto.

### 2.2 Synoptic history of Dorian and Humberto

Dorian originated from a tropical wave that moved off the west coast of Africa on August 19, 2019. The wave crossed the tropical Atlantic becoming a tropical depression on August 24 at 0600 UTC at about 1296 km east-southeast of Barbados and a tropical storm at 1800 UTC that day. Dorian's intensity increased further while moving west-northwest reaching hurricane category by September 27 and major hurricane category by September 30, while being centered about 713 km east of the northwestern Bahamas (Figure 1A). Dorian became a category 5 hurricane on the Saffir-Simpson Hurricane Scale in the northwestern Bahamas on September 1 with estimated winds of 296 km h$^{-1}$. At this time, it moved very slowly westward making landfall on Grand Bahama Island on September 2 with winds of 287 km h$^{-1}$ (Figure 1A). Dorian remained stationary over this area weakening to category 4 and 3 (Figure 1A) because of its interaction with land and the induced-ocean cooling. Then, it

turned northwestward and started moving along the east of Florida during the period from 3–5 September as a category 2, but as its core moved over the Gulf Stream, Dorian regained strength and reached category 3 again offshore of the coasts of Georgia and South Carolina (Figure 1A). Afterwards, Dorian made landfall over Cape Hatteras on September 6 with 157 km h$^{-1}$ and left the Sargasso Sea this day, when moving beyond 35°N (Avila et al., 2020).

On the other hand, Humberto primarily originated from a weak tropical wave that started off the west coast of Africa on August 27 and moved westward across the tropical Atlantic approaching the Lesser Antilles on September 4. By September 13 it became a tropical depression east of the central Bahamian island of Eleuthera at 1800 UTC and 6 hours later it became a tropical storm (Figure 1A). Then, Humberto turned northwestward and maintained that direction for the next 48 hours at a slow translation speed. It reached hurricane category on September 16 about 278 km east-northeast of Cape Canaveral, Florida (Figure 1A). At this point, Humberto started to move east-northeastward strengthening to a major hurricane on September 18 (Figure 1A). Thereafter, Humberto weakened steadily and became an extratropical cyclone by September 20 (Stewart, 2020).

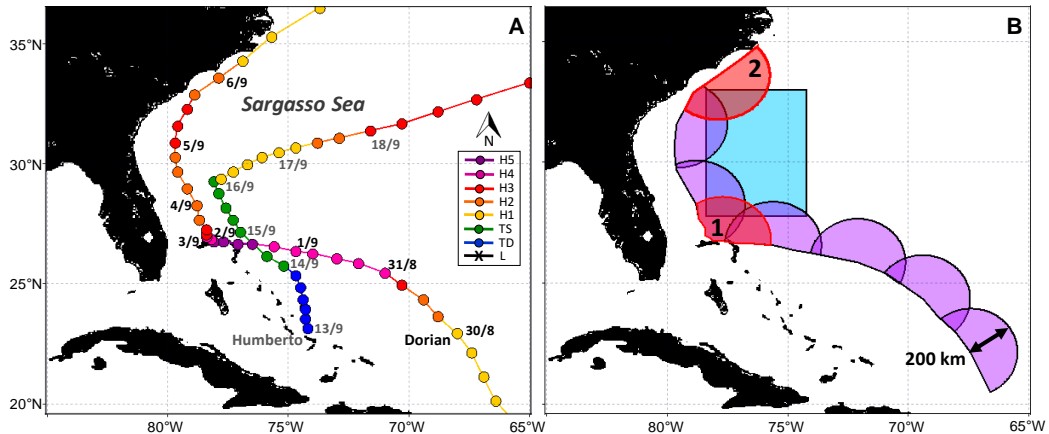

**Figure 1.** (A) Trajectory of tropical cyclones (TCs) Dorian and Humberto (2019). Colours indicate the TC category (i.e., L: Low-pressure system, TD: Tropical depression, TS: Tropical storm, H1–H5: Saffir Simpson Hurricane Categories). Numbers along trajectories indicate the day/month. (B) 200 km semi-disks along Dorian's trajectory and the square area in the western Sargasso Sea. Red semi-disks 1 and 2 indicate the areas where chlorophyll-a concentration profiles were analysed.

## 2.3 Methodology

We analysed the oceanic response along the trajectory of Dorian in 200 km radius semi-disks centered at consecutive hurricane positions in the 20–35°N latitudinal band (limits of the Sargasso Sea) (Figure 1B). We followed this asymmetric approach due to the fact that Dorian moved close to the east coast and shelf waters of the United States of America (USA), so, the waters to the left side of its trajectory can be considered as optically complex waters (i.e., Case 2 waters of the classification of Morel, 1980), which limit the use of ocean colour data. On the other hand, coastal/shelf and oceanic waters respond differently to the passage of TCs (Zhao et al., 2015). For instance, coastal areas are strongly affected by freshwater discharge and flooding lead-

ing to a high concentration of nutrients and post-storm phytoplankton productivity (Farfán et al., 2014; Paerl et al., 2019). Thus, in order to account for the oceanic response in the Sargasso Sea, we restricted to the deep waters to the right of Dorian's trajectory. Given that upwelling, cooling and deepening of the isothermal layer are more intense on the right side of a TC's trajectory in the Northern Hemisphere (Price, 1981; Hanshaw et al., 2008; Gierach and Subrahmanyam, 2008; Fu et al., 2014), our asymmetric approach captures the strongest oceanic response. Besides these Lagrangian measurements along Dorian's trajectory, we also investigated the oceanic response in a square area (Eulerian) located to the right of its trajectory (Figure 1B), where a considerable post-storm sea surface cooling was observed. The strongest oceanic response accounting for the combined effect of Dorian and Humberto was observed in this square area since it was affected by the intensive winds near the center of Dorian and it was crossed by Humberto. In contrast, the entire area along Dorian's trajectory was affected by Humberto to a lesser extent, especially the waters surrounding Grand Bahama Island since Humberto crossed the semi-disk 1 (see Figure 1). Moreover, storm surge and saltwater inundation were reported along the east coast of Florida following the passage of Humberto (Stewart, 2020), indicating that this area was also exposed to its forces.

We assessed the daily response of the oceanographic variables before, during and after the passage of Dorian. We considered the pre-storm week (i.e., days –10 to –3 before hurricane passage) as a benchmark for comparison with the four post-storm weeks (i.e., from day 0 to 30, where day 0 refers to the day the hurricane entered the study area) in agreement with previous studies (Menkes et al., 2016; Avila-Alonso et al., 2019). For what concerns the square area, we considered day 0 as the day that Dorian started to impact it, i.e., as soon as there was overlap of the 200 km radius semi-disks and the square polygons. Overall, the first two post-storm weeks account for Dorian-induced oceanic effects, while the third and fourth post-storm weeks account for the combined effects of both Dorian and Humberto. Daily arithmetic means of the studied variables were computed along Dorian's trajectory and in the square study area. For the former, the mean values of the consecutive semi-disks were averaged to retrieve the daily mean along the entire hurricane trajectory, in agreement with Babin et al. (2004).

## 2.4 Data

### 2.4.1 Response variables

We considered SST and chl-a concentration as the main physical and biological response variables, respectively, as in previous studies in the region (e.g., Babin et al., 2004; Shropshire et al., 2016). SST data were derived from the Operational SST and Sea Ice Analysis (OSTIA) Near Real Time level 4 product (Donlon et al., 2012), provided by the Copernicus Marine Environment Monitoring Service (CMEMS, *http://marine.copernicus.eu*). OSTIA merges both infrared and microwave radiometer data, together with in situ observations at a spatial resolution of $0.05° \times 0.05°$.

We used the CMEMS GlobColour multisatellite merged data of near real-time chl-a concentration (level 4 cloud free product) which is based on a spatial and temporal interpolation of the level 3 product at a spatial resolution of $0.0417° \times 0.0417°$ (Garnesson et al., 2019b, a). The chl-a analyses involve multiple chl-a algorithms, i.e., CI algorithm for oligotrophic waters (Hu et al., 2012) and OC5 algorithm for mesotrophic and coastal waters (Gohin et al., 2002; Gohin, 2011). The CI and OC5 continuity is ensured using the same approach as that utilized by NASA. When the chl-a concentration is in the range from 0.15 to

0.2 mg m$^{-3}$, a linear interpolation is used (Garnesson et al., 2019b). The algorithm validation has shown a good relationship between in situ measurements and satellite observations of chl-a concentration, while daily level 4 products in general show a low bias (0.04) and a coefficient of determination of 0.71 at global scale (Garnesson et al., 2019a).

Moreover, we also investigated the subsurface oceanographic variability by analysing data of the OHC, which corresponds the integrated thermal energy from the sea surface to 26 °C isotherm depth (Leipper and Volgenau, 1972; Price, 2009), and the average temperature from the sea surface to a depth of 100 m (T$_{100}$, a typical depth of vertical mixing by a category 3 hurricane) (Price, 2009). The latter has been suggested as an adequate variable to identify subsurface thermal fields (Price, 2009), so we can draw sound conclusions on the subsurface cooling as a consequence of the TCs. Similarly, we analyzed chl-a concentration profiles to assess the post-storm biological response throughout the euphotic zone (0–200 m).

We used daily OHC data from the Systematically Merged Regional Atlantic Temperature and Salinity (SMARTS) Climatology adjusted to a two-layer reduced gravity model at a spatial resolution of 0.25° × 0.25° (Meyers et al., 2014) (data available at *ftp://ftp.nodc.noaa.gov/pub/data.nodc/sohcs*). SMARTS blends temperature and salinity fields from the World Ocean Atlas 2001 (WOA01) and the Generalized Digital Environmental Model (GDEM) version 3.0 based on their performance compared to in situ measurements (Meyers et al., 2014). SMARTS estimations of OHC during hurricane seasons show little bias and low normalized root mean squared difference (RMSD normalized by the mean of the local in situ observations) in the North Atlantic basin in general, with the lowest values in the western Sargasso Sea (e.g., RMSD≈0.3) (see Figure 2-16 in Shay et al., 2019). Temperature profiles were derived from the Global Ocean Physics Analysis and Forecasting product at a spatial resolution of 0.083° × 0.083°, provided by Mercator Ocean and distributed by CMEMS. This product uses version 3.1 of the NEMO (Nucleus for European Modelling of the Ocean) ocean model and has 50 vertical levels (22 levels within the upper 100 m) with a decreasing resolution from 1 m at the sea surface to 450 m into the deep ocean (Lellouche et al., 2016).

Chl-a profiles were obtained from the CMEMS Global Biogeochemical Analysis and Forecast product of Mercator Ocean. This product is a global biogeochemical simulation result (at 0.25° × 0.25° spatial resolution) obtained using the PISCES (Pelagic Interactions Scheme for Carbon and Ecosystem Studies) model which is part of the NEMO model (Lamouroux et al., 2019). The vertical grid of this product has 50 levels (ranging from 0 to 5500 m), with a resolution of 1 m near the sea surface and 400 m into the deep ocean (Lamouroux et al., 2019). Such chl-a concentration profiles have shown good agreement with both satellite and in situ measurements. Moreover, the depth of the DCM is quite accurately simulated for the North Atlantic subtropical gyre (Lamouroux et al., 2019).

### 2.4.2 Drivers of ocean cooling

Post-storm SST cooling is a very complex process, involving hydrodynamic (vertical mixing and upwelling) and thermodynamic (surface heat flux) processes (Vincent et al., 2012a). Taking into account that cooling as a consequence of surface heat loss is more relevant towards the shallow waters of the continental shelf (Morey et al., 2006; Ezer, 2018; Wei et al., 2018), we assessed the variability of the mixed layer and the thermocline displacement induced by Dorian and Humberto in order to identify the drivers of the post-storm-induced cooling. The mixed layer depth (MLD) data were derived from the Global Ocean Physics Analysis and Forecasting product distributed by CMEMS. MLD is defined by sigma theta considering a variable

threshold criterion (equivalent to a 0.2 °C decrease), i.e., the depth where the density increase compared to density at 10 m depth corresponds to a temperature decrease of 0.2 °C in local surface conditions (Chune et al., 2019).

Upwelling and downwelling regimes can be identified by analysing the fluctuations in 20 °C isotherm depth (D20) (Jaimes and Shay, 2009, 2015), since D20 generally occurs within the area of maximum vertical temperature gradient in the tropical ocean. Therefore, D20 is considered as a proxy for the thermocline (Delcroix, 1984; Reverdin et al., 1986; Seager et al., 2019).

The subtropical western Sargasso Sea has a particular vertical structure of its water column with a permanent thermocline at approximately 400–500 m depth (Hill, 2005) and a seasonal thermocline in the upper ocean in summer (Hatcher and Battey, 2011). Between the seasonal and permanent thermocline, there is a layer with relatively homogeneous conditions, known as the western North Atlantic subtropical mode (STMW) water or 18-degree water because of its typical temperature (Schroeder et al., 1959; Kwon and Riser, 2004; Stevens et al., 2020). This layer is often defined by a temperature range of 17–19 °C (Billheimer

and Talley, 2016), though a recent study used a broader range (16–20 °C) in order to account for mesoscale variability and recent warming of STMW (Stevens et al., 2020). Consequently, we consider that the analysis of the upwelling response in terms of fluctuations in D20 will give insights into the vertical displacement of cool waters from the boundary of the seasonal thermocline and the STMW layer. We inferred daily D20 data from the SMARTS Climatology. D20 estimates have shown low RMSD value (RMSD≈0.2) during hurricane seasons in the western Sargasso Sea (see Figure 2-16 in Shay et al., 2019).

Overall, the oceanic response induced by a TC is determined by a combination of both atmospheric and oceanic variables (Babin et al., 2004). Although we did not directly analyse the impact of atmospheric variables such as wind speed and rainfall on the upper ocean response, we consider that through the assessment of the MLD and D20 variability the effects of the above-mentioned atmospheric variables are taken into account. It has been suggested that wind-driven processes (i.e., vertical mixing and upwelling) govern the post-storm SST cooling in the waters surrounding Cuba due to the

high and statistically significant correlation of wind speed and SST, as well as the consistent temporal variability of wind speed, MLD, D20 and SST anomalies during and after the passage of a hurricane (Avila-Alonso et al., 2019, 2020). Moreover, model simulations have revealed that TC rainfall can reduce the post-storm MLD, thus reducing cold water entrainment (Jacob and Koblinsky, 2007; Jourdain et al., 2013). Overall, fresh rainwater affects vertical stability and therefore modulates ocean mixing (Jacob and Koblinsky, 2007; Jourdain et al., 2013).

Given the heterogeneous nature of the datasets used in this study, i.e., satellite and modelled data obtained using different methods at different spatial resolutions, we focus on highlighting general spatio-temporal patterns of the analysed variables, since some specific dynamics may be a consequence of this heterogeneity.

## 3 Results

### 3.1 Physical response

#### 3.1.1 Ocean cooling

During the pre-storm week of Dorian, the spatially averaged SST values along its trajectory and in the square study area were 29.6 and 29.4 °C, respectively, which were high compared to climatological records for the period 1982–2018, i.e., 29.02 and 28.73 °C, respectively. This points to a weak thermal contrast between the Gulf Stream Current and the adjacent waters to its right (Figure 2A). Then, after the passage of Dorian, a considerable surface cooling was observed to the right side of its trajectory during the first and second post-storm weeks (Figure 2B and C). During the pre-storm week of Dorian, TC Erin affected the northerwestern Sargasso Sea (Blake, 2019) leading to a moderate SST cooling in the first post-storm week of Dorian (Figure 2A and B). Then, after the passage of Humberto at the end of the second post-storm week, an intensive surface cooling was observed in the entire area (Figure 2D and E). At the end of the third post-storm week and beginning of the fourth one, TC Jerry moved across the central northwestern Atlantic basin as a tropical storm and then weakened to a low-pressure system (Brown, 2019) (Figure 2E). In Figure 2E we can see a patch of considerable low SSTs to the left of Jerry's trajectory (centered at 31 °N and 70 °W approximately), which could have resulted from the combined effects induced by Humberto and Jerry. This patch of low SSTs was located to the right of Humberto's trajectory, who affected this area as a category 3 hurricane (Figure 1A) a week before the passage of Jerry.

When analysing the daily evolution of SST anomalies, we found that both along the trajectory and in the square study area, SST started to decrease 3 days before Dorian's arrival reaching a maximum cooling of approximately 1 °C 1 to 3 days after its passage (Figure 3A). The storm-induced subsurface thermal variability (described by OHC and $T_{100}$) changed in phase with SST (Figure 3) though OHC and $T_{100}$ differed significantly (Mann-Whitney test, $p < 0.05$) between the studied areas (Figure 3B and C). More specifically, OHC and $T_{100}$ values were 11 and 2% smaller (more negative) along Dorian's trajectory as compared with the square study area during the first post-storm week (Figure 3B and C). Then, during the third and fourth post-storm weeks SST, OHC and $T_{100}$ decreased, with the lowest anomalies in the square study area (Figure 3). In general, SST, OHC and $T_{100}$ anomalies were then 34, 28 and 42% lower in this area than along Dorian's trajectory. Furthermore, we found that during the first/second and third/fourth post-storm weeks 45 and 90% of SST anomalies in the square study area accounted for a surface cooling of more than 1 °C, respectively, while 4 and 47% of SST anomalies accounted for a surface cooling of more than 2 °C, respectively (Figure 4). Overall, SST, OHC and $T_{100}$ anomalies in the square study area were 50, 63 and 57% lower in the third/fourth post-storm weeks than in the first/second post-storm weeks, respectively.

#### 3.1.2 Drivers of ocean cooling

We found fluctuations of the MLD before, during and after the passage of Dorian with no significant differences between the studied areas when analysing the entire post-storm period (Mann-Whitney test, $p > 0.05$). The mixed layer started to deepen 3 days before Dorian's arrival in both study areas, reaching a maximum mean deepening of 10 m along Dorian's trajectory at day

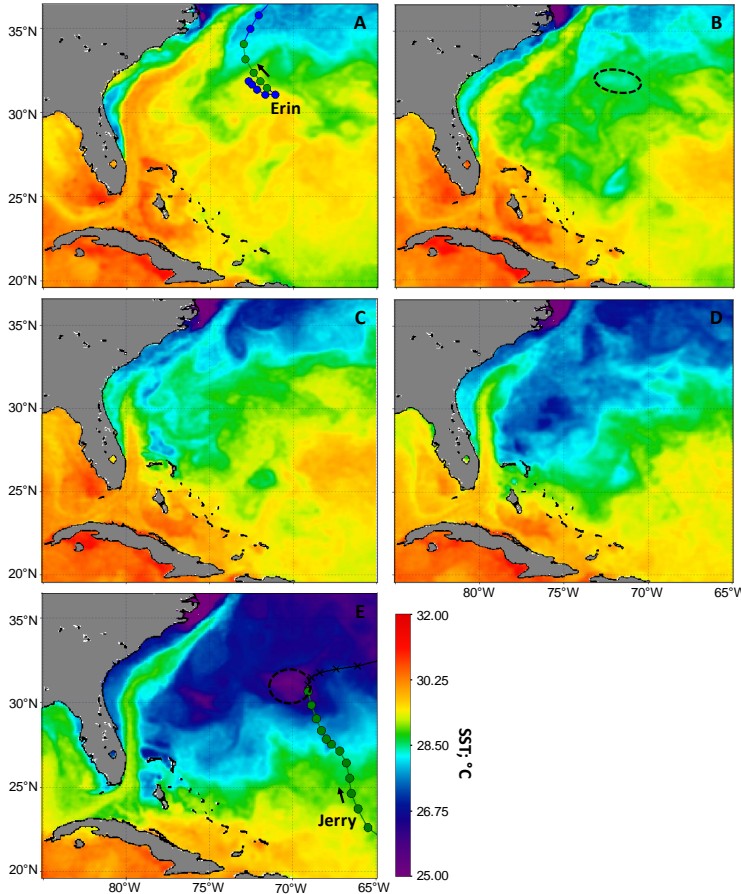

**Figure 2.** Weekly mean sea surface temperature (SST) in the (A) pre-storm week and (B) first, (C) second, (D) third and (E) fourth post-storm weeks of Dorian in the Sargasso Sea. The trajectories of Erin and Jerry are superimposed on (A) and (E), respectively, with colour coding as defined in Figure 1A and arrows indicating their forward movement. The dashed contours in (B) and (E) indicate the probable surface cooling induced by Erin and Jerry, respectively.

1 and 11 m in the square study area at day 2 (Figure 5A). A second maximum deepening was observed at the end of the second post-storm week and the beginning of the third one (Figure 5A). On the other hand, D20 fluctuated considerably during the analysed post-storm period, with significantly different values between the studied areas (Mann-Whitney test, $p < 0.05$). The largest upward displacement of D20 during the first two post-storm weeks along the trajectory (Figure 5B) was determined by the majority of pixels accounting for shoaling of this isotherm (61 and 56% of pixels along the trajectory and in the square

area, respectively, Figure 5C) since in both areas the net displacement of D20 was similar at this time (Figure 5D). This spatial pattern changed during the third and fourth post-storm weeks, with a higher upward displacement of D20 in the square study area (Figure 5B), resulting from both the majority of pixels accounting for shoaling of this isotherm (60 and 73% of pixels

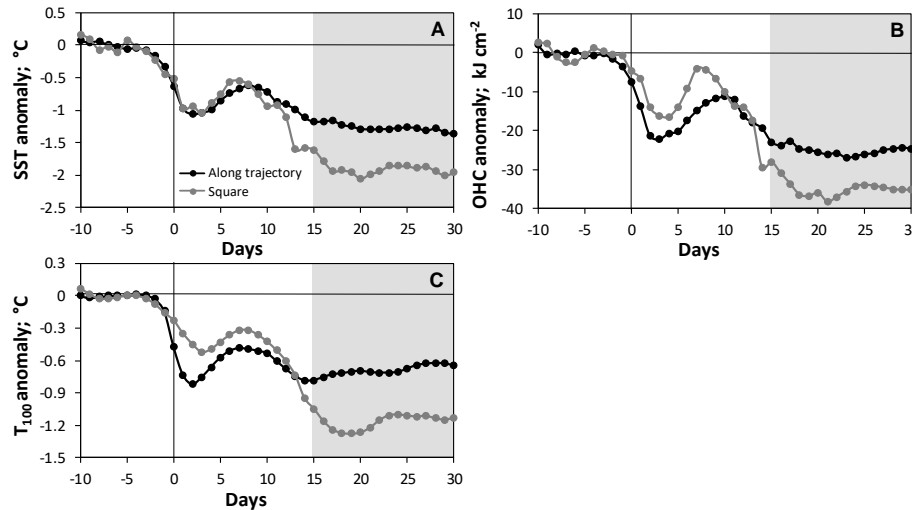

**Figure 3.** Daily mean evolution of anomalies of (A) sea surface temperature (SST), (B) ocean heat content (OHC) and (C) average temperature from the sea surface to a depth of 100 m ($T_{100}$) in the Sargasso Sea before, during and after the passage of Dorian and Humberto. The grey shaded area depicts the third and fourth post-storm weeks of Dorian which account for the combined effects of Dorian and Humberto.

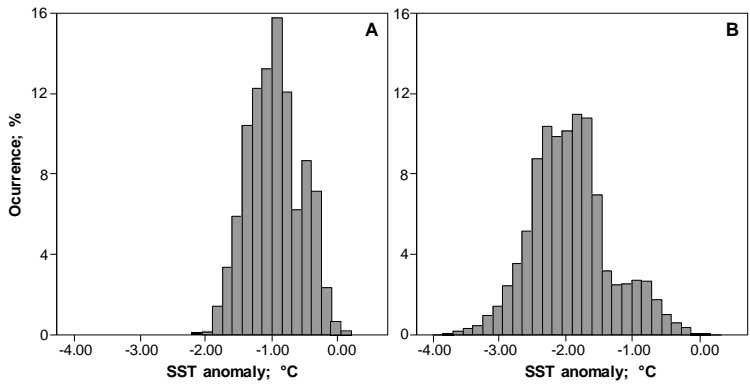

**Figure 4.** Sea surface temperature (SST) anomalies in the (A) first/second and (B) third/fourth post-storm weeks of Dorian in the square study area shown in Figure 1B.

along the trajectory and in the square area, respectively, Figure 5C) and a higher net displacement of D20 in those pixels (40 and 45 m along the trajectory and in the square, respectively, Figure 5D).

### 3.1.3 Climatological analysis of sea surface temperature

In order to assess the magnitude of the physical and biological response induced by Dorian and Humberto in the square study area, we compared the mean SST and chl-a concentration during the third and fourth post-storm weeks of Dorian with the

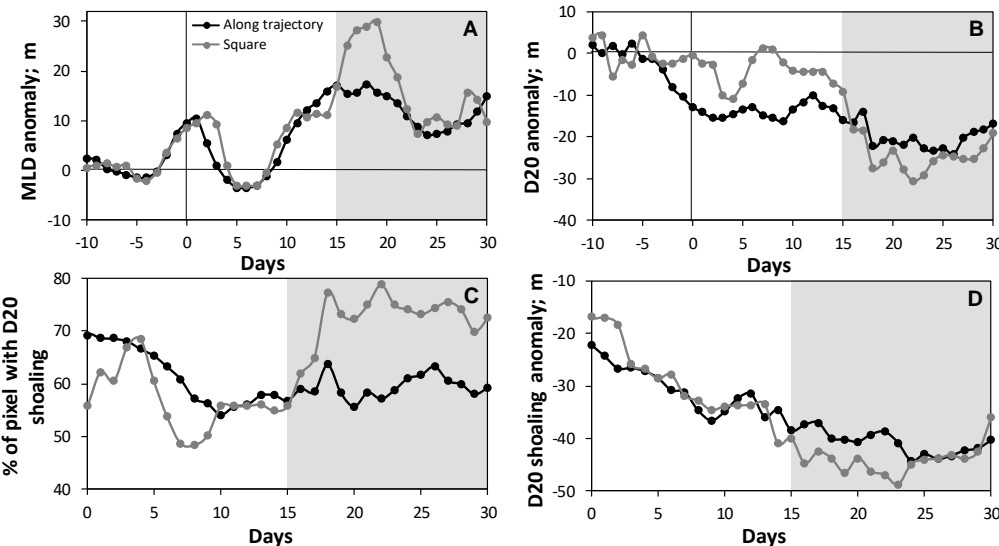

**Figure 5.** Daily mean evolution of anomalies of (A) mixed layer depth (MLD) and (B) 20 °C isotherm depth (D20) before, during and after the passage of Dorian and Humberto in the Sargasso Sea. (C) Percentage of pixels within each study area indicating upwelling in the post-storm period. From those pixels with upwelling, (D) displays the magnitude of the upward displacement of D20 as a measure of the seasonal thermocline upwelling. The grey shaded area depicts the third and fourth post-storm weeks of Dorian which account for the combined effects of Dorian and Humberto.

climatological records. The actual third and fourth post-storm weeks of Dorian in this area were the ones of 19–26 September 2019 and 27 September–4 October 2019, respectively, so, we assessed the SST and chl-a concentration variability during these

250 weeks in the previous years for which satellite observations were available, i.e., 1982–2018 for SST and 1998–2018 for chl-a concentration. We refer to these weeks as the *third and fourth post-storm weeks of Dorian*, regardless of the analysed year, and to indicate the actual oceanic response induced by Dorian and Humberto we specified this was the one in 2019. We computed weekly and corresponding long-term arithmetic means (i.e., climatologies) for each variable within the square study area, while standardized anomalies (deviations from the climatological weekly mean) were calculated by subtracting the long-term weekly

mean from the weekly mean of each variable. From Figure 6A we can see that the SST anomaly in the *third post-storm week of Dorian* in 2019 was the most negative one in the last 14 years, while only 18% of the analysed years showed anomalies lower than the one in 2019 during this week. On the other hand, the SST anomaly in the *fourth post-storm week of Dorian* in 2019 was the most negative one in the last 20 years, while only 13% of the analysed years showed anomalies lower than the one in 2019 (Figure 6A).

Despite the confirmed cooling induced by Dorian and Humberto, a positive trend can be observed in the time series of SST anomalies (Figure 6A), which could bias the results stated above. This positive trend is consistent with the reported global surface warming and specifically in the northwestern Atlantic Ocean (Bulgin et al., 2020). In order to compare 2019 SST anomalies with the ones in the previous years on the same basis, i.e., without the effect of ocean warming in the region, we

removed the linear trend. Thus, we calculated the least squares regression line for this time series and subtracted the deviations from this line (i.e., the difference between the real observed value and the residual one). From Figure 6B we can see that when removing the positive trend in data, SST anomalies in 2019 were the coldest ones in the climatological record except for the ones in 1984 and 1999. In general, SST values in the analysed weeks in 2019 were significantly lower (Mann-Whitney test, $p < 0.05$) than the climatological values at this time (Figure 7A). When assessing the SST response induced by Dorian and Humberto in a spatially explicit way, we observed that, indeed, these TCs led to a substantial SST cooling over the western Sargasso Sea (Figure 6C and D).

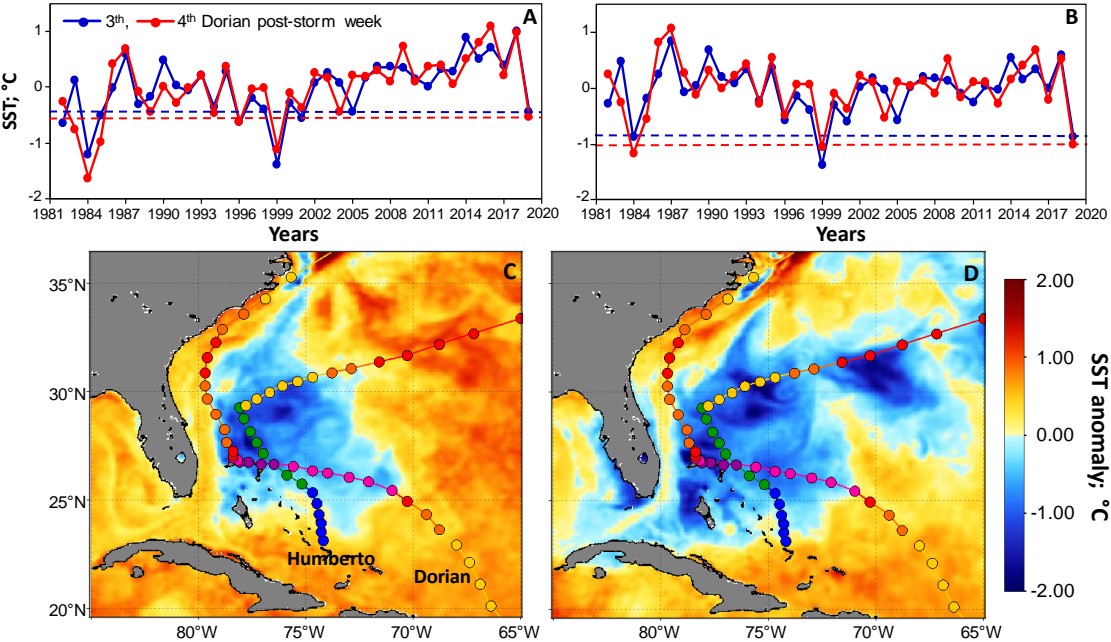

**Figure 6.** (A) Time series of weekly mean sea surface temperature (SST) anomalies during the *third and fourth post-storm weeks of Dorian* in the square study area shown in Figure 1B. (B) Detrended time series of SST anomalies. The dashed lines mark anomalies in 2019 for comparison with the previous years. (C) and (D) Spatially explicit SST anomalies (Dorian+Humberto (2019) induced effects – Climatology (1982–2018)) in the *third and fourth post-storm weeks of Dorian*, respectively.

## 3.2    Biological response

### 3.2.1    Chlorophyll-a concentration bloom

During the pre-storm week of Dorian, the spatially averaged chl-a concentration along its trajectory and in the square study area were 0.074 and 0.049 mg m$^{-3}$, respectively, which were low compared to climatological records for the period 1998–2018, i.e., 0.088 and 0.060 mg m$^{-3}$, respectively. Overall, chl-a concentration was low and spatially homogeneous in the deep waters

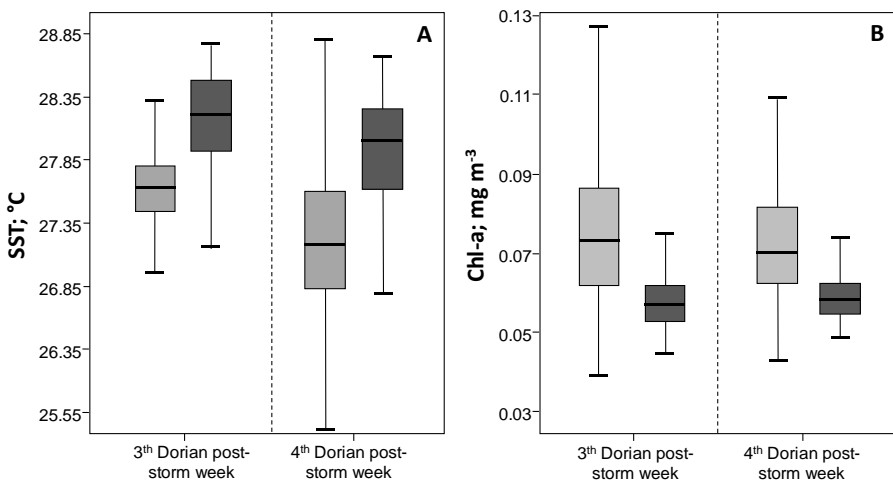

**Figure 7.** Dorian and Humberto induced (A) sea surface temperature (SST) and (B) chlorophyll-a (chl-a) concentration variability (light gray) with climatological records (dark grey) during the *third and fourth post-storm week of Dorian* in the square study area shown in Figure 1B.

of the western Sargasso Sea at this time, being quite distinct from the high-chl-a band along the east coast of USA (delineated by the 0.1 mg m$^{-3}$ contour in Figure 8A). Then, following the passage of Dorian, we observed an oceanic chl-a increase and also an expansion of the coastal high-chl-a band during the entire post-storm period (Figure 8B–E). The oceanic biological response was strongest in the waters to the north and northeast of Grand Bahama Island, especially during the third and fourth post-storm weeks (Figure 8D and E).

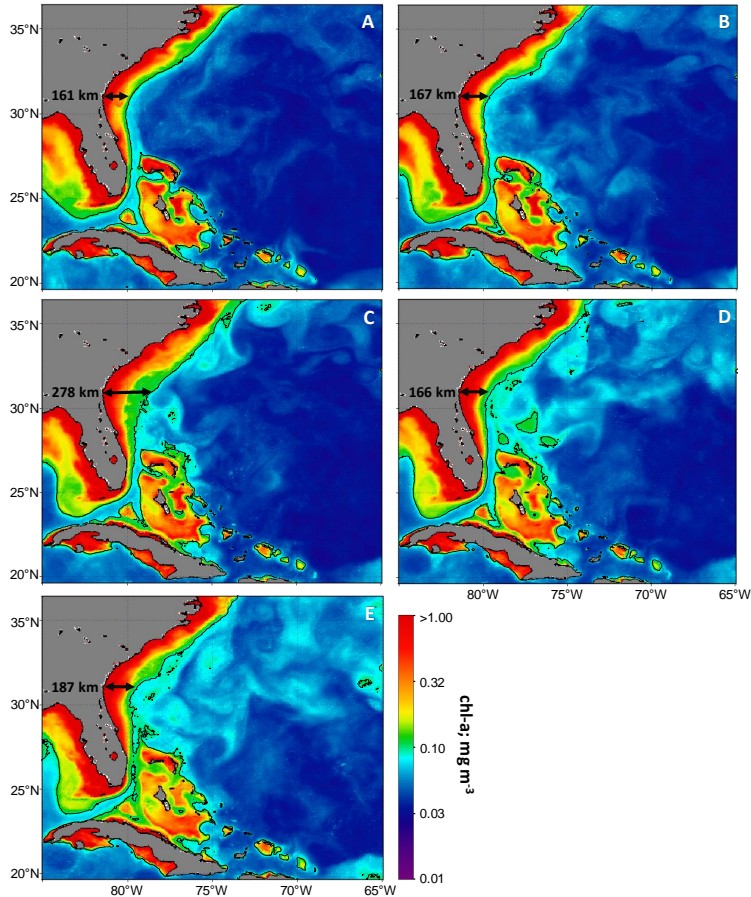

**Figure 8.** Weekly mean chlorophyll-a (chl-a) concentration in the (A) pre-storm week and (B) first, (C) second, (D) third and (E) fourth post-storm weeks of Dorian in the Sargasso Sea. The contour lines delineate chl-a concentration values 0.1 mg m$^{-3}$ apart. Arrows indicate the distance from the coast to the 0.1 mg m$^{-3}$ chl-a contour.

Taking into account the daily evolution of chl-a concentration anomalies we found that chl-a started to increase 3 days before Dorian's arrival (only along its trajectory, Figure 9A). Then, chl-a peaked at days 2 and 3 along its trajectory and in the square study area, respectively, with anomalies being 23% higher in the former area than in the latter one during the first two post-storm weeks. A second chl-a maximum was observed at the end of the second post-storm week along the trajectory, while in the square study area, chl-a concentration fluctuated considerably from the end of the second post-storm week on but still showing higher anomalies than the ones along the trajectory at this time (Figure 9A). When analysing the frequency of occurrence of chl-a concentration anomalies in the square study area during the first/second and third/fourth post-storm weeks, we found that 55 and 80% of the anomalies were higher than 0.01 mg m$^{-3}$, respectively, while, 36 and 57% of the anomalies were higher than 0.02 mg m$^{-3}$, respectively (Figure 9B and C). Overall, surface chl-a concentration anomalies were 16% higher in the third/fourth post-storm weeks than in the first/second post-storm weeks in this area.

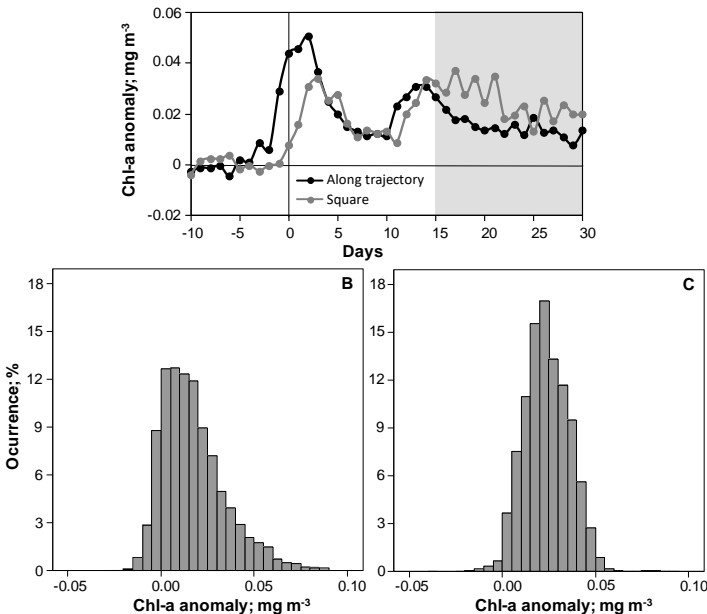

**Figure 9.** Daily mean evolution of (A) chlorophyll-a (chl-a) concentration anomalies in the western Sargasso Sea before, during and after the passage of Dorian and Humberto. The grey shaded area depicts the third and fourth post-storm weeks of Dorian which account for the combined effects of Dorian and Humberto. Distributions of chl-a concentration anomalies in the (B) first/second and (C) third/fourth post-storm weeks of Dorian in the square area delineated in Figure 1B.

In order to assess the subsurface biological response, we analysed vertical profiles of chl-a concentration in the square study area and in semi-disks 1 and 2 (see Figure 1B) given the high post-storm spatial variability of chl-a concentration profiles along Dorian's trajectory. Semi-disk 1 was crossed by both studied TCs, while semi-disk 2 was only affected by Dorian. Although the DCM does not always coincide with biomass or productivity maxima, it often corresponds to peaks in abundance of phytoplankton (reviewed by Moeller et al., 2019). Thus, we considered the chl-a concentration in the DCM (chl-a$_{\text{DCM}}$) as a proxy of phytoplankton abundance. Figure 10A shows that although the depth of the DCM (DCM$_{\text{Z}}$) did not change in the post-storm period, the mean chl-a concentration in the euphotic zone (chl-a$_{200}$) and chl-a$_{\text{DCM}}$ were 4 and 16% higher in the third/fourth post-storm weeks than in the first/second post-storm weeks, respectively (Figure 10A and Table 1). Besides, the chl-a profiles in semi-disk 1 showed the highest post-storm variability (Figure 10B). In this area, chl-a$_{200}$ and chl-a$_{\text{DCM}}$ were 9 and 16% higher in the third/fourth post-storm weeks than in the first post-storm week, respectively (Table 1). Moreover, DCM$_{\text{Z}}$ was shallower in most post-storm weeks as compared to the pre-storm week (Figure 10B and Table 1), while in the second post-storm week the DCM could not be discerned (Figure 10B). Overall, the strong biological response in the third/fourth post-storm weeks as compared to the first/second post-storm weeks in these two areas is consistent with the maximum deepening of the MLD and shoaling of D20 (Figure 10A and B and Table 1). In contrast, the chl-a profiles in

semi-disk 2 evolved differently (Figure 10C) with chl-a$_{200}$ being 25% lower in the third/fourth post-storm weeks than in the
first/second post-storm weeks (Figure 10C and Table 1).

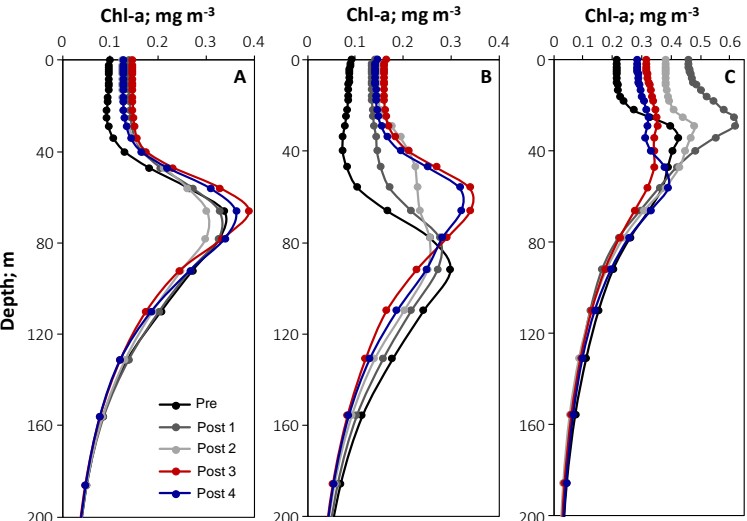

**Figure 10.** Spatially averaged chlorophyll-a (chl-a) concentration profiles in the (A) square study area and the 200 km radius semi-disks (B)
1 and (C) 2 shown in Figure 1B.

### 3.2.2   Climatological analysis of surface chlorophyll-a concentration

From Figure 11A we can infer that the mean chl-a concentration anomaly in the *third post-storm week of Dorian* in 2019
was the highest one in the last 9 years, while only 23% of the analysed years showed anomalies higher than the one in 2019.
On the other hand, the chl-a concentration anomaly in the *fourth post-storm week* in 2019 was only surpassed by the one
in 1999 (Figure 11A). In general, the chl-a concentration increase induced by the combined effect of Dorian and Humberto in
the square area was significantly higher (Mann-Whitney test, $p < 0.05$) than climatological records (Figure 7B). In agreement
with the spatial distribution of the most negative anomalies of SST shown in Figure 6C, the most positive chl-a anomalies
during the *third post-storm week* were observed to the north and northeast of Grand Bahama Island (Figure 11B) affected
by the eyewall winds of these TCs. Conversely, in the *fourth post-storm week* positive chl-a anomalies (and negative SST
anomalies in Figure 11C) were more evenly spread although, in general, a considerable response occurred in the latitudinal
band 25–30 °N (Figure 11C).

**Table 1.** Mean chlorophyll-a concentration in the euphotic zone (chl-a$_{200}$, mg m$^{-3}$), chl-a concentration at the deep chlorophyll maximum (chl-a$_{DCM}$, mg m$^{-3}$), depth of the DCM (DCM$_Z$, m), mixed layer depth (MLD, m) and 20 °C isotherm (D20, m) in the square study area, and semi-disks 1 and 2 shown in Figure 1B during the pre- and post-storm weeks of Dorian.

| | Pre | Post 1 | Post 2 | Post 3 | Post 4 |
|---|---|---|---|---|---|
| **Square** | | | | | |
| chl-a$_{200}$ | 0.132 | 0.159 | 0.149 | 0.167 | 0.155 |
| chl-a$_{DCM}$ | 0.338 | 0.328 | 0.301 | 0.389 | 0.362 |
| DCM$_Z$ | 66 | 66 | 66 | 66 | 66 |
| MLD | 14 | 18 | 23 | 36 | 25 |
| D20 | 198 | 193 | 193 | 172 | 173 |
| **Semi-disk 1** | | | | | |
| chl-a$_{200}$ | 0.111 | 0.148 | 0.161 | 0.175 | 0.164 |
| chl-a$_{DCM}$ | 0.299 | 0.276 | - | 0.339 | 0.319 |
| DCM$_Z$ | 92 | 78 | - | 56 | 56 |
| MLD | 15 | 20 | 21 | 32 | 27 |
| D20 | 253 | 223 | 225 | 209 | 211 |
| **Semi-disk 2** | | | | | |
| chl-a$_{200}$ | 0.237 | 0.386 | 0.322 | 0.270 | 0.260 |
| chl-a$_{DCM}$ | 0.426 | 0.618 | 0.479 | - | 0.390 |
| DCM$_Z$ | 34 | 29 | 29 | - | 56 |
| MLD | 20 | 17 | 33 | 27 | 30 |
| D20 | 180 | 185 | 177 | 189 | 209 |

## 4 Discussion

### 4.1 Ocean cooling and its drivers

The extensive surface cooling observed to the right of Dorian's trajectory (Figure 2B and C) was also reported by Ezer (2020). Besides, this extensive cooling was consistent with the one observed after the passage of Hurricane Matthew (2016) (who followed a similar trajectory as Dorian) and results from the interaction between a hurricane and the core of the Gulf Stream flow and its associated eddy field (Ezer, 2018). Dorian induced a disruption of the Gulf Stream flow, disconnecting the upstream Florida Current from the downstream Gulf Stream and weakening the Gulf Stream flow by almost 50% (Ezer, 2020). This direct effect on surface currents was followed by intense surface cooling largely determined by the reduced flow of warm tropical waters being advected downstream Gulf Stream as well as mixing of the upper oceanic layer (like during the Hurricane Matthew) (Ezer et al., 2017; Ezer, 2020). Overall, the interaction of hurricanes with the Gulf Stream induces a considerable vertical mixing and reduction of the stratification frequency (Kourafalou et al., 2016). Consequently, it has been reported that

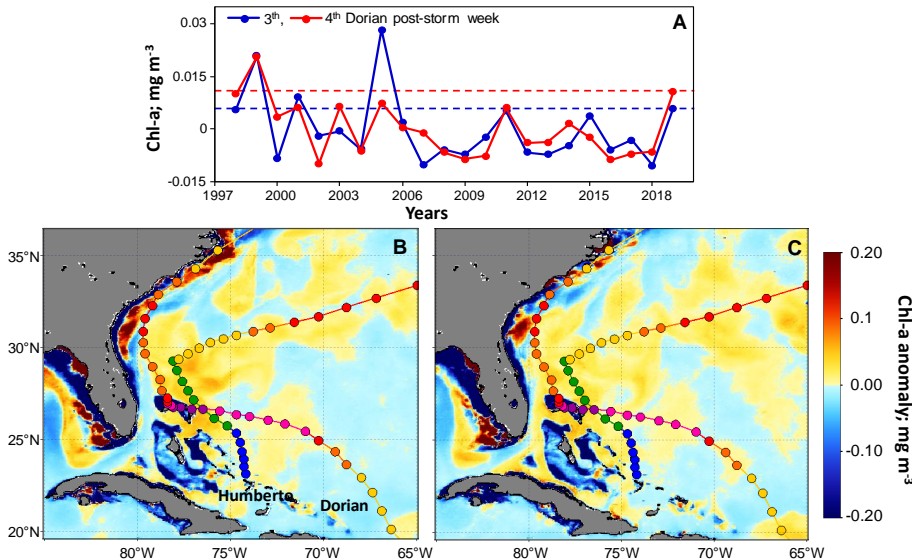

**Figure 11.** (A) Time series of weekly mean chlorophyll-a (chl-a) concentration anomalies during the *third and fourth post-storm weeks of Dorian* in the square study area delineated in Figure 1B. The dashed lines mark anomalies in 2019 for comparison with the previous years. (B) and (C) Spatially explicit chl-a anomalies (Dorian+Humberto induced effects (2019) – Climatology (1998–2018)) in the *third and fourth post-storm weeks of Dorian*, respectively.

vertical mixing drove cooling of the upper 50 m after the passage of Matthew across the Sargasso Sea (Ezer, 2018). On the other hand, we found that in some cases, maximum surface cooling occurred over areas with low values of sea surface height (SSH) and oceanic cyclonic circulation, which correspond to cold-core eddies (Figure 12A and C). More specifically, the extensive cooling observed in the center of the basin in Figure 12A largely agrees with the low SSH values in it (Figure 12C). Owing to the special thermodynamic structure of cyclonic eddies, the MLD within them is relatively shallower than in the adjacent ocean, which reinforces the uplifting of cold and nutrient (chl-a-)rich subsurface waters by TC-induced vertical mixing (Ning et al., 2019). This also explains the increased chl-a concentration to the right of Dorian's trajectory in regions with oceanic cyclonic circulation (Figure 12B and C).

The fact that SST anomalies in both study areas are similar until day 11 (Figure 3A) follows for the fact that the center of Dorian moved over the warm waters of the Gulf Stream current. Despite the Dorian-induced SST decrease in the area, still the warm core of the Gulf Stream (although weakened) was observed most post-storm days (e.g., Figures 2B and 12A). On the other hand, it has been reported that sea surface cooling induced by TCs can extend over vast areas far from the area affected by the TC center because of the remotely-induced mixing (Oey et al., 2006, 2007; Vincent et al., 2012a; Menkes et al., 2016; Ezer, 2018). This contributed to the SST decrease in the square study area and the deepening of the mixed layer 3 days before the arrival of Dorian in both study areas (Figures 3A and 5A). Summer conditions in the Sargasso Sea are characterized by a strong thermal stratification of the upper water column leading to shallow mixed layers (< 20 m) (Michaels et al., 1993; Steinberg

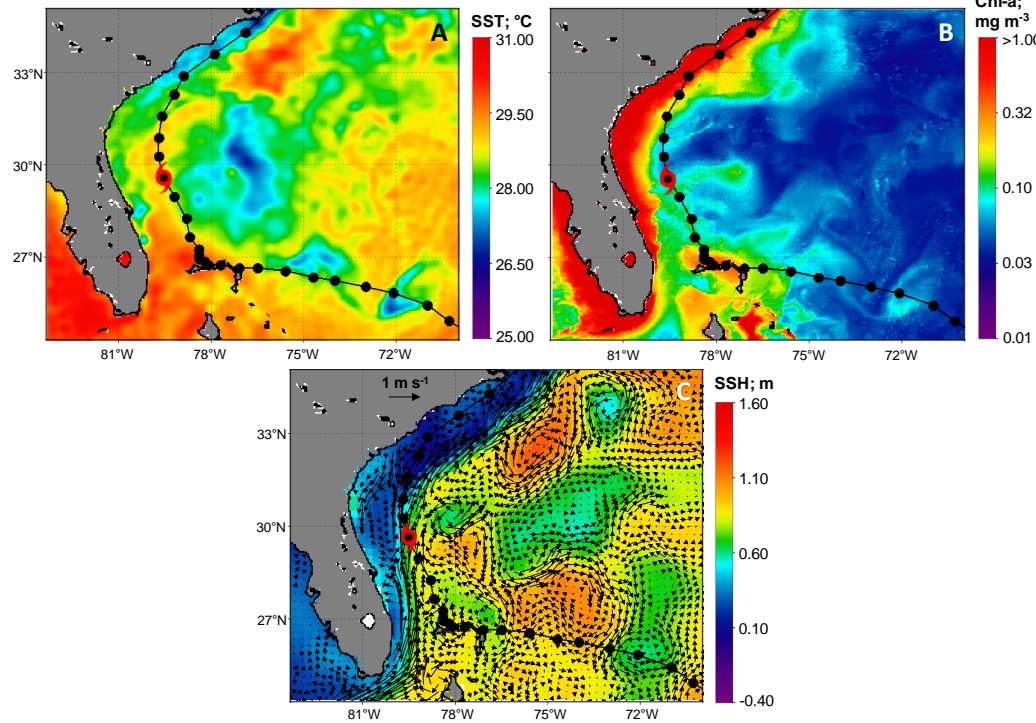

**Figure 12.** Oceanic response to Dorian (September 4, 2019). (A) Sea surface temperature (SST), (B) chlorophyll-a (chl-a) concentration and (C) absolute sea surface height (SSH) with the absolute geostrophic velocity vectors superimposed (data derived from Salto/DUACS gridded multimission altimeter at $0.25° \times 0.25°$ spatial resolution). The trajectory of Dorian and a hurricane symbol indicating its estimated position is superimposed in all imagines.

et al., 2001). Spatially averaged MLD during the pre-storm week were approximately 14 and 16 m along the trajectory and in the square study area, respectively. Shallow mixed layers indicate that there is cold water near the surface, so that wind-induced mixing generates immediate surface cooling. This agrees with the fact waters of the western Sargasso Sea have a low resistance to TC-induced cooling through mixing (see Figure 5b in Vincent et al., 2012b).

     The retrieved temporal dynamics of the MLD during the first two post-storms weeks (Figure 5A) largely agrees with the one
reported by Foltz et al. (2015). These authors found that during and immediately following a cyclone's passage, the spatially averaged MLD in the Sargasso Sea increases sharply by 7 m on average, while the positive anomaly rapidly diminishes during the subsequent days. Then, after the post-storm shoaling of the MLD, it can oscillate (shoaling/deepening) around the same mean depth (Foltz et al., 2015; Zhang et al., 2016; Prakash et al., 2018). However, this temporal evolution considers the oceanic response to an individual TC. The second maximum deepening observed in our study at the beginning of the third post-storm
355    week (Figure 5A) agrees with the suggestion made by Ezer et al. (2017), stating that several storms affecting the same region within a relatively short period of time have a cumulative impact on ocean mixing.

For what concerns the deep cooling induced by Dorian and Humberto (Figure 3B and C), we consider this was largely attributed to upwelling of the thermocline, which explains the more negative anomalies of OHC and $T_{100}$ along the trajectory in the first two post-storm weeks and in the square study area during the last two post-weeks (Figure 3B and C). TCs typically give rise to upwelling flow underneath the storm centre and weak downwelling of the displaced warm water over a broad area beyond the upwelled regions (Price, 1981; Jullien et al., 2012; Fu et al., 2014). Overall, TC-induced upwelling by Ekman pumping plays a major role in inducing cooling under the TC center (e.g., Jullien et al., 2012; Wei et al., 2018).

### 4.2 Chlorophyll-a concentration bloom

The increased post-storm chl-$a_{200}$ and chl-$a_{DCM}$, as well as the upward displacement of the DCM (Figure 10 and Table 1), is consistent with previous observations in several oceanic basins around the world (e.g., Ye et al., 2013; Chacko, 2017; Chakraborty et al., 2018; Jayaram et al., 2019). A DCM typically occurs in stratified and oligotrophic marine environments where phytoplankton are not evenly distributed throughout the water column (Liccardo et al., 2013; Macías et al., 2013). In this case, the shape of the vertical profiles of chl-a concentration can be considered as Gaussian according to the classification of Mignot et al. (2011, 2014). In contrast, in environments with a deep and strong mixing, phytoplankton are homogeneously distributed in the mixed layer and chl-a profiles are sigmoid (Mignot et al., 2011, 2014). Given the TC-enhanced vertical mixing and deepening of the mixed layer, a shift from Gaussian to sigmoid-like shape of the chl-a profiles can be expected after the passage of TCs across a stratified area (Wu et al., 2007). However, a considerable post-storm change of chlorophyll in the surface mixed layer occurs when the MLD is deeper than the DCM (Wu et al., 2007), which explains the Gaussian-shaped profiles after the passage of Dorian and Humberto (Figure 10) since MLD was typically shallower than the DCM (Table 1).

On the other hand, the increased post-storm chl-$a_{200}$ (Figure 10 and Table 1) indicates that subsurface phytoplankton bloom was due to new production resulting from a nutrient influx into the euphotic layer instead of a vertical chl-a redistribution. This also explains the increased post-storm chl-$a_{DCM}$ as well as the shallower $DCM_Z$ (Figure 10 and Table 1). The formation and persistence of the DCM is governed by biophysical processes accounting for both bottom-up and top-down controls (Cullen, 2015; Moeller et al., 2019). However, given that light and nutrient availability are primary factors governing phytoplankton abundance and distribution, they have also been recognized as relevant drivers of phytoplankton communities in the DCM (Latasa et al., 2016). It has been reported that turbulence (like the one induced by storms) can enlarge the phytoplankton density above the DCM (Liccardo et al., 2013) leading to a shoaling of the DCM. Given that the DCM occurs at the transition between the light-limited region (below the maximum) and nutrient-limited region (above the maximum), a limited addition of nutrients into the region immediately above the DCM, shifts the system from a nutrient-limited regime to a light-limited one (Liccardo et al., 2013).

Although strong mixing due to TCs is mostly confined to the upper ocean, the cyclone-induced inertial waves affect the subsurface mixing too (Prakash et al., 2018). Part of the energy transferred to inertial currents by the storm may propagate below the mixed layer and finally drive turbulent mixing in the thermocline (Cuypers et al., 2013). Given that wind-generated inertial waves can propagate up to 2000 m depth in the Sargasso Sea (Morozov and Velarde, 2008), this subsurface mixing can reach the DCM and nitracline and consequently transport chl-a and nutrients to the mixed layer (Foltz et al., 2015). The

nitracline depth in the northern Sargasso Sea in summer oscillates between 90–150 m (Malone et al., 1993; Goericke and Welschmeyer, 1998), though it decreases after the passage of TCs (Malone et al., 1993). So, the upward nutrient advection from the nitracline after the passage of TCs can fuel subsurface phytoplankton productivity.

For what concerns the post-storm surface chl-a response, we argue that it was mainly determined by entrainment of chl-a from deep waters given the rapid increase of the surface chl-a concentration as soon as a TC reached the study areas, which agrees with the suggestion of Shropshire et al. (2016). Figure 12B shows a patch of high chl-a concentration (0.13 mg m$^{-3}$) to the right of Dorian's trajectory the day it affected the area confirming the immediate biological response to the TC forcing. Although the mixed layer did not reach the DCM in the study area (Table 1), it is likely that a small amount of chl-a was eventually transported to the surface since deep near-inertial mixing can raise the surface chl-a concentration (Wang et al., 2020). The higher chl-a concentration anomalies along the trajectory as compared the ones in the square study area from day – 3 to 3 (Figure 9A) are associated with both an increased upward displacement of D20 (Figure 5B) and the horizontal advection of chl-a-rich waters from the coast of the northwestern Bahamas islands and the east coast of USA, which were impacted by Dorian (Avila et al., 2020). These higher chl-a concentration anomalies are consistent with the fact that surface phytoplankton blooms are common near the TC trajectory (e.g., Babin et al., 2004; Walker et al., 2005; Gierach and Subrahmanyam, 2008; Lin and Oey, 2016; Avila-Alonso et al., 2019).

### 4.3 Climatological analysis of sea surface temperature and chlorophyll-a concentration

TC activity in the North Atlantic basin peaks from late August to September (Neely, 2016). Thus, given that the third and fourth post-storm weeks of Dorian occurred mainly in September, the time series of weekly mean anomalies of SST and chl-a concentration (Figures 6A and 11A) indicated the oceanic responses to preceding TCs. For instance, extreme anomalies observed in 1999 (Figures 6A and 11A) were associated with the increased oceanic response induced by the passage of consecutive Hurricanes Dennis and Floyd at the end of August and September 1999 as will be shown in the next section (Figure 13D and F). Other extreme SST and chl-a concentration anomalies were the ones in 1984 and 2005 (Figures 6A and 11A) associated with Hurricanes Diana (1984) and Ophelia (2005). Both hurricanes followed peculiar trajectories since they made clockwise loops leading to a prolonged forcing time over the ocean and consequently to a strong oceanic response (Lawrence and Clark, 1985; Beven and Cobb, 2006).

The strong oceanic response to the combined effect of Dorian and Humberto was reinforced by the fact that they were slow-moving TCs. We computed the translation speed of Dorian and Humberto following the procedure outlined by Babin et al. (2004) and Gierach and Subrahmanyam (2008) using the "best track" observations of time and position from the HURDAT2 database of the National Hurricane Center (*http://www.aoml.noaa.gov/hrd/ hurdat/hurdat2.html*). In order to classify these TCs on the basis of their translation speed we followed the approach by Avila-Alonso et al. (2020), i.e., a fast (slow) moving TC has a higher (lower) translation speed than the climatological one computed for Atlantic TCs averaged in 5-degree latitude bins (see table in *https://www.aoml.noaa.gov/hrd-faq/#tropical-cyclone-climatology*, consulted in May 2020). We found that Dorian and Humberto had a mean translation speed of 10.95 and 12.8 km h$^{-1}$, respectively, in the 25–30 °N latitudinal band, while the corresponding mean climatological translation speed is 20.1 km h$^{-1}$. This finding agrees with the decreasing TC translation

speed observed at global scale in general, and in the North Atlantic basin in particular, as a consequence of the global-warming-induced changes in global atmospheric circulation (Kossin, 2018; Lanzante, 2019; Moon et al., 2019; Yamaguchi et al., 2020). Although we acknowledge that pre-storm oceanographic conditions influence the magnitude of the induced-oceanic post-storm response (Nigam et al., 2019), our climatological analysis revealed that the strongest oceanographic response in the western Sargasso Sea is associated with consecutive TCs and long-lasting TC forcing.

## 4.4 Temporal evolution of the oceanic response to consecutive TCs

Climatological SST responses to the passage of TCs in all TC-prone regions around the world indicate that maximum cooling occurs 2 days after the TC passage and then cooling starts decreasing from this day onward (Menkes et al., 2016). Moreover, although the TC-induced cold wake in the North Atlantic basin needs about 60 days to disappear, it decays (80%) in the first 20 days (Haakman et al., 2019). On the other hand, even though post-storm blooms can last about 2–3 weeks, their peak occurs following the passage of TCs (at days 3 to 4) (Menkes et al., 2016). However, we found that the passage of a second TC disrupts these global patterns leading to a decreasing SST trend and a second chl-a bloom. It could be thought that the oceanic disturbances induced by Dorian would have limited the impacts from Humberto. The latter TC affected the study area approximately two weeks after the passage of Dorian, so, there was a short time period for the ocean to fully recover from the Dorian-induced variability. It has been reported that the TC-induced oceanic variability can limit the sea surface cooling and deepening of the mixed layer induced by a second TC (Baranowski et al., 2014). These authors found a SST decrease of 0.67 °C after the passage of Typhoon Hagupit, while the second Typhoon Jangmi (passing 7 days after Hagupit) only induced a decrease of 0.22 °C.

In order to assess the individual oceanic response induced by Humberto and to compare it with the one induced by Dorian, we computed spatially averaged SST and chl-a anomalies in semi-disk 1 since this area was affected for both studied TCs (see Figure 1). We considered the oceanic pre-conditions to the passage of Dorian and Humberto (i.e., days –10 to –3 before Dorian and Humberto arrival on September 2 and 14, respectively, at semi-disk 1) as a benchmark for comparison with its corresponding post-storm weeks as was described in Section 2.3. We found that Dorian and Humberto induced a spatially averaged mean SST decrease of –1.1 and –0.6 °C, respectively, during their first post-storm week in semi-disk 1 (results not shown). The weak cooling induced by Humberto can be attributed to the oceanic conditions left by Dorian and also due to the strength of Humberto since it affected the study area as a tropical storm (see Figure 1). However, despite the limited surface cooling induced by Humberto, the mean SST anomaly during the third post-storm week of Dorian (accounting for the combined effects of Dorian and Humberto) was –1.95 °C in semi-disk 1. This finding agrees with the one of Baranowski et al. (2014) since the minimum SST value during the entire analysed post-storm period was observed after the passage of the second typhoon (see Figure 8 in Baranowski et al., 2014) indicating that the passage of consecutive TCs within a short period of time superimposes effects on the ocean.

In order to confirm the increased oceanic response induced by the combined effects of consecutive TCs in the western Sargasso Sea, we compared the temporal evolution of SST and chl-a concentration anomalies induced by Dorian and Humberto with the ones induced by an individual hurricane and other consecutive hurricanes. Thus, we assessed the oceanic response in

200 km radius semi-disks in the latitudinal band 20–35 °N to individual Hurricane Irene (2011) and consecutive Hurricanes Dennis and Floyd (1999) (which were two weeks apart approximately (Pasch et al., 1999; Beven, 2000)) since they followed a similar trajectory as Dorian (Figure 13A and B). Irene passed across the optically shallow waters of The Bahamas where reflectance of the seafloor contributes to the reflected light measured by satellites (Dierssen et al., 2010) leading to data quality issues when quantifying, for instance, chl-a. Thus, we discarded extreme observations of chl-a concentration in this area. Although it has been reported that Dorian induced a similar Gulf Stream Current variability and surface cooling as Matthew since they followed similar trajectories (Ezer, 2020), the latter passed closer to the east coast of USA than the former (Stewart, 2017). Thus, we did not assess the oceanic response to Matthew (as individual hurricane) since the retrieved oceanic variability would be highly influenced by coastal processes.

Overall, we found that maximum sea surface cooling and chl-a concentration increase induced by Irene occurred immediately after its passage and then cooling and phytoplankton bloom started to decrease from this day onward (Figure 13C and E) in agreement with the global climatological responses to a TC passage (Menkes et al., 2016). The slight decrease of SST and the increase of chl-a concentration at the end of the third post-storm week of Irene and beginning of the fourth one were essentially determined by the remote impact of Hurricane Katia who passed more than 600 km from the area affected by the center of Irene (Figure 13A). In contrast, when analysing the oceanic response to consecutive Hurricanes Dennis and Floyd, we observe that the most intense cooling and phytoplankton bloom occurred at the end of the third post-storm week (i.e., after the passage of Floyd) (Figure 13D and F), which is consistent with the temporal evolution of SST and chl-a concentration anomalies retrieved for Dorian and Humberto (Figures 3A and 9A). It has been reported that consecutive TCs within a short period of time superimpose effects on the upper ocean response because of strong-induced mixing and upwelling (Wu and Li, 2018; Ning et al., 2019; Wang et al., 2020) as was observed after the passage of Dorian and Humberto (Figure 5). Furthermore, in our study area, the passage of multiple hurricanes superimposes effects on the ocean leading to a reduced Gulf Stream volume transport (Todd et al., 2018; Ezer, 2020), which in turn impacts surface cooling.

The TC-induced oceanographic variability impacts marine trophic communities in general (e.g., Fiedler et al., 2013; Hung et al., 2014). Pedrosa-Pàmies et al. (2018, 2019) reported that Hurricane Nicole (2016) caused phytoplankton bloom and increased zooplankton and bacterial abundance in the deep water column in the Bermuda Time-Series Site in the Sargasso Sea, indicating a coupled post-storm response of primary and secondary production. The Sargasso Sea is the spawning area of the European and American eels (*Anguilla anguilla* Linnaeus 1758 and *A. rostrate* LeSueur, respectively) (Schmidt, 1923; van Ginneken and Maes, 2005; Friedland et al., 2007; Miller et al., 2019a). Spawning and retention of larvae are associated with relatively productive zones that may enhance feeding opportunities (Kleckner and McCleave, 1988; Munk et al., 2010). Eels survival during their early larvae stages is strongly correlated with food availability, specifically with planktonic organisms (Riemann et al., 2010; Ayala et al., 2018). Consequently, climate-driven processes affecting nutrient availability and phytoplankton primary production affect levels of eel larval survival in areas with high spawning activity, as well as its recruitment (Bonhommeau et al., 2008a, b; Miller et al., 2016). Considering that the TC-induced chl-a concentration changes may impact the survival rates of fish larvae and their recruitment to adulthood in the Sargasso Sea (Foltz et al., 2015), the

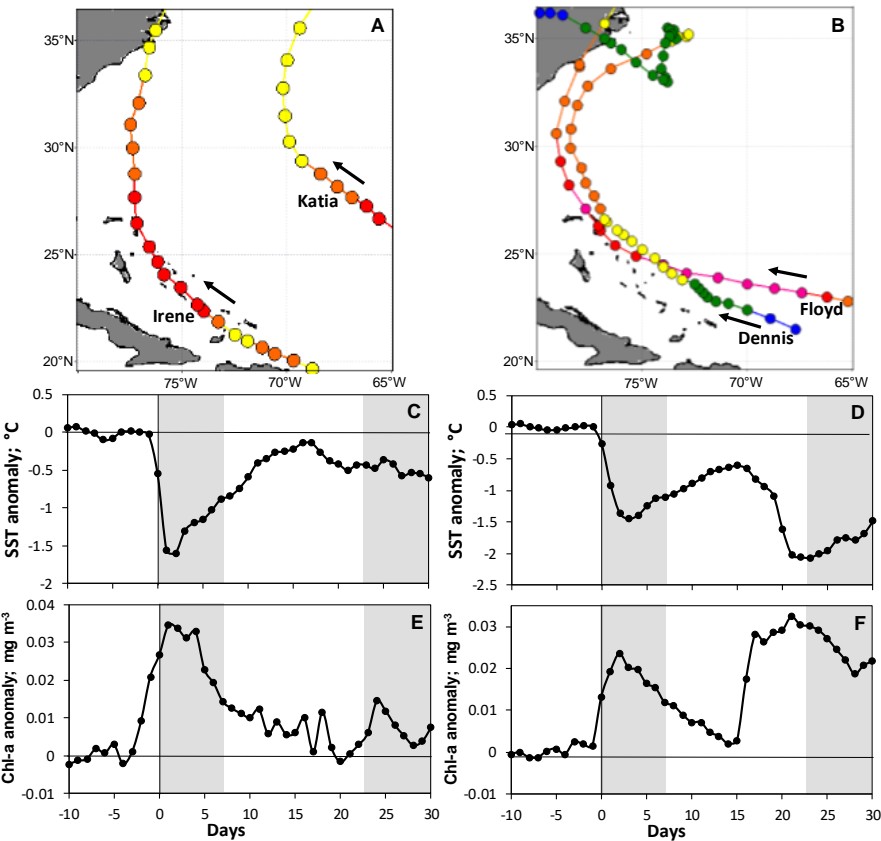

**Figure 13.** Trajectories of Hurricanes (A) Irene and Katia (2011) and (B) Dennis and Floyd (1999), with colour coding as defined in Figure 1A and arrows indicating their forward movement. Daily mean evolution of anomalies of (C and D) sea surface temperature (SST) and (E and F) chlorophyll-a (chl-a) concentration induced by Irene (C and E) and Dennis+Floyd (D and F). The grey shaded area depicts the first and fourth post-storm weeks.

assessment of the oceanic response to TCs serves for future studies addressing the influence of climate variability on fishery oceanography in the region.

## 5  Conclusions

In this work, multi-platform datasets were used to investigate the physical and biological responses of the upper ocean to the consecutive Hurricanes Dorian and Humberto in the western Sargasso Sea. The consistency of our results with literature confirms the suitability of the datasets used to capture the post-storm response in the study area. In general, the main finding of this work is the additional oceanic variability (i.e., increased ocean cooling and chl-a concentration, deepening of the mixed layer and enhanced upwelling) observed after the passage of a second TC in the western Sargasso Sea indicating that

consecutive TCs within a short period of time superimpose effects on the upper ocean response. More specifically, Dorian induced an extensive surface cooling to the right side of its trajectory which was associated with the presence of cold-core eddies in the area. This extensive cooling was mainly driven by vertical mixing while the post-storm decreases of OHC and $T_{100}$ were mostly governed by the upward displacement of D20. On the other hand, Dorian led to an increased chl-a concentration,

which was 23% higher along the trajectory than in the square study area during the first two post-storm weeks. This biological response is essentially determined by entrainment of chl-a from depth and horizontal advection of coastal productive waters. Then, at the end of the second post-storm week of Dorian, Humberto started to affect the study area superimposing effects on the upper ocean response because of the strong-induced mixing and upwelling. Overall, anomalies of SST, OHC and $T_{100}$ were 50, 63 and 57% lower in the third/fourth post-storm weeks than in the first/second post-storm weeks of Dorian, respectively,

while surface chl-a concentration anomalies, chl-a$_{200}$ and chl-a$_{DCM}$ were 16, 4 and 16% higher in the third/fourth post-storm weeks than in the first/second post-storm weeks, respectively. Furthermore, the combined effects of these TCs induced an upward displacement of the DCM in some analysed cases. This subsurface biological response indicates the stimulation of new production because of the upward nutrient fluxes. The surface SST and chl-a concentration responses induced by Dorian and Humberto were significantly more intense (Mann-Whitney test $p < 0.05$) as compared to climatological records.

Our climatological analysis revealed that the strongest TC-induced oceanographic variability in the western Sargasso Sea has been associated with consecutive TCs and long-lasting TC forcing. Nevertheless, further studies are needed to improve our understanding of the relationship between consecutive TCs and their interaction with the ocean in order to derive the general behavior of the upper ocean response to consecutive TCs in the western Sargasso as well as in other oceanic basins. Overall, the oceanic response to the passage of Dorian and Humberto reported in this study gives insights into the oceanic implications

of a simultaneous increase of both the frequency and intensity of TCs in the North Atlantic basin. Thus, together with a future increase of TC activity in the region, an increased oceanic response could also be expected. Moreover, considering that the TC-induced chl-a concentration changes may impact the survival rates of fish larvae and their recruitment to adulthood in the Sargasso Sea, the results presented here serve as a basis for future studies addressing the influence of climate variability on fishery oceanography in the region.

*Data availability.* For satellite products presented in this work, please contact the corresponding author at davila@uclv.cu

*Author contributions.* All authors together have contributed to the research reported in different ways. D.A.-A., conceived and designed this study in general, conducted the collection, processing and analysis of data and wrote the original draft. J.M.B., contributed to methodological aspects and to the design of the study, carefully reviewed and edited the English. R.C., assisted to the interpretation of results, and provided comments and suggestions on the paper. Revising it for important intellectual content. B.D.B carefully reviewed and edited the English.

Provided comments and suggestions on the paper. Supervision and funding acquisition.

*Competing interests.* The authors declare that they have no known competing financial interests or personal relationships that could have appeared to influence the work reported in this paper.

*Acknowledgements.* This work was supported by the Special Research Fund (BOF) of Ghent University, Belgium, Grant Code 01W03715.

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
