# Peer review of "Oceanic response to the consecutive Hurricanes Dorian and Humberto (2019) in the Sargasso Sea"

_Natural Hazards and Earth System Sciences, 2020_

## Referee Comment (RC1) · Anonymous Referee #1 · 17 Nov 2020

This study examines the upper-ocean response to the passage of Dorian and then Humberto in the same area about two weeks later. Humberto caused significant additional deepening of the mixed layer, shoaling of the thermocline, cooling of SST, and increase in chl-a concentration.

The manuscript is well written and organized, with plenty of statistics and assessments of statistical significance. The passage of two hurricanes over the same region within two weeks of each other is uncommon, so it's a good case study for analysis.

My main criticism is that it's not clear what is new or surprising about the results or why they are important. These aspects need to be emphasized, and I think additional

analysis will be needed to do it. It seems like the unexpected or surprising result might be that there was so much additional increase in MLD, decrease in SST, increase in chl-a after Humberto, even though Dorian caused significant changes beforehand. One might have thought that the preconditioning of Dorian would have suppressed the impacts from Humberto. Is the strength of the ocean's response to Humberto due to the quick ocean recovery after Dorian passed? Or was Humberto strong enough to cause even more cooling and enhanced chl-a in the well-mixed upper-ocean that Dorian left behind? Topics such as these should be investigated and discussed (and maybe comparisons made to other single or multiple TC passages in the western Atlantic), otherwise readers will not know how to interpret the results or why they might be important.

Minor comments:

Lines 203-204: It's unclear what 'respectively' refers to because you mention deepening of 10 and 11 m, along trajectory and in the square area, and days 1 and 2. Is it deepening of 10 and 11 m along the trajectory and in the square area, respectively? Or at day 1 and 2, respectively?

Lines 220-225: It looks like there's a large positive SST trend in Fig. 6A. What if you remove the trend before calculating the 2019 percentages? Do they change?

Line 375: Several papers dispute the result of Kossin (2018). These should also be cited (Lanzante, Uncertainties in tropical-cyclone translation speed. Nature., 570, E6–E15, 2019; Yamaguchi et al., Nature Communications, 11, 47, 2020; Moon et al., Climate change and tropical cyclone trend, Nature, 570, E3–E5, 2019)

---

## Referee Comment (RC2) · Anonymous Referee #2 · 20 Nov 2020

General Comments:

1. The study describes the response of the ocean to two consecutive hurricanes in 2019, looking at various aspects, including temperature change, heat content and chlorophyll. The paper is well written and interesting, though the results are not unexpected or especially new. It should be noted though that studies of consecutive hurricanes in this region are not new - the authors may have not been aware for example of some recent studies (that should be cited) of the impact of consecutive hurricanes on the same region, including Todd et al. (2018) on the 2017's hurricanes and Ezer (2020) who studied the same Dorian and Humberto hurricanes of this study! However,

the aforementioned papers focused mostly on the impact of hurricanes on coastal sea level and ocean currents (and the Gulf Stream in particular), while here the additional analysis of biological impacts and chlorophyll is an interesting new kink that worth publication. With regards to ocean dynamics, it would be useful to add a little more discussion - the study area is strongly influenced by the Gulf Stream which was significantly weakened during Dorian (Ezer, 2020) and this had implications for mixing and advection, since a weakened GS could also contribute to relative cooling downstream when advection of warm tropical waters slows down. It would have been interesting also to extend the study period to see the timescale of recovery after the storm and the role that the GS plays in this recovery.

References:

Todd, R.E., T.G. Asher, J. Heiderich, J.M. Bane, R.A. Luettich (2018), Transient response of the Gulf stream to multiple hurricanes in 2017, Geophys. Res. Lett., 45, doi:10.1029/2018GL079180.

Ezer, T. (2020), The long-term and far-reaching impact of hurricane Dorian (2019) on the Gulf Stream and the coast, J. Mar. Sys., 208, doi:10.1016/j.jmarsys.2020.103370.

Specific Comments:

2. Abstract- Sentence starting "Overall, ..." (line 7) and ending "...post-storm weeks, respectively." (line 12) is too long, cumbersome and needs separating to shorter sentences and rephrasing.

3. Introduction- The discussion is mostly about climate change and hurricanes, and not enough on the actual topic of the study, i.e., the processes involved in the impact of storms on the upper ocean.

4. Line 35- I suggest deleting "In agreement", its awkward to start a paragraph this way.

5. Lines 38-39- Maybe add citation to Todd et al. (2018) who studied the same mentioned hurricanes, and when discussing Dorian (lines 42, 62, etc.) may cite Ezer (2020).

6. Figures 6a and 11a- It is unclear how they were done and what they represent- how can one compare a hurricane at a particular year and a particular week to past years?, and what does the clear trend represents. It seems that there is much more information in these figures than described and the interpretation is unclear.

7. Lines 305-312- When discussing the dynamics and mixing processes, may be add something about the role of the Gulf Stream, as in Ezer (2020).

8. Fig. 12c- Are those absolute geostrophic velocity? or velocity anomaly? Anomalies do not tell us much, maybe also show the absolute SSH and velocity to indicate the hurricane track relative to the mean flow of the subtropical gyre and the GS.

---

## Referee Comment (RC3) · Jill Trepanier (Referee) · 1 Dec 2020

This paper focuses on the oceanic responses to the passing of two hurricanes in 2019. It is well-written, topically relevant to the current literature, and showcases some interesting graphics. I have read through the previous reviewers' reports so I will add new information below and potentially reinforce some of the early reviews.

I agree that the introduction focuses more on climate change and TCs and less on upper ocean response. Please add literature and introductory comments on the expectation of upper ocean changes from the passing of hurricanes. An additional paragraph should suffice. Do not take out what you have - it is well written and supports the ideas.

You need only add a little more. This will help the remaining parts of your paper pack more of an important "punch" if we are more aware of why it matters to be looking at Chl-a.

Section 2.1. Please refer to Figure 1 as your "study area graphic", it will lead to your figure being moved up slightly which helps the reader identify the location.

Did you do any assessment of other conditions of the atmosphere in the "pre-storm" week of Dorian? Meaning, did it rain at all during that time? Rainfall from minor to severe thunderstorms can alter your SSTs in the region and, without mention of it, I'm not sure that other atmospheric events might be biasing your results. You mention TC Erin. Were there any other events? Please comment.

Paragraph including line 190: the way you worded the beginning of this paragraph is confusing. Please rephrase.

Great Figure 2. Really showcases what you are describing.

In paragraph with line 275, you bring it up yourself that the findings are similar with Ezer (2018) (or later, Foltz et al. 2015). So why is your study different? And, thus, why is it important that it is published? I think you need a stronger argument than you have presented here. Broader impacts related to your study can be helpful here.

I think your conclusion needs to be strengthened. While you provide a nice summary of what are interesting statistics, you need to relate it to a broader picture. Why does knowing this information help us in some way? Does it inform fisheries? Does it inform management practices? Why does it matter? (I believe it does, but you need to provide a stronger argument for it).

---

## Author Comment (AC1) · 11 Dec 2020

The reviewer is thanked for his/her positive and constructive comments on the manuscript.

**REVIEWER 1**

**My main criticism is that it's not clear what is new or surprising about the results or why they are important. These aspects need to be emphasized, and I think additional analysis will be needed to do it. It seems like the unexpected or surprising result might be that there was so much additional increase in MLD, decrease in SST, increase in chl-a after Humberto, even though Dorian caused significant changes beforehand. One might have thought that the preconditioning of Dorian would have suppressed the impacts from Humberto. Is the strength of the ocean's response to Humberto due to the quick ocean recovery after Dorian passed? Or was Humberto strong enough to cause even more cooling and enhanced chl-a in the well-mixed upper-ocean that Dorian left behind? Topics such as these should be investigated and discussed (and maybe comparisons made to other single or multiple TC passages in the western Atlantic), otherwise readers will not know how to interpret the results or why they might be important.**

In the new version of the manuscript we have included new information in order to emphasize the novelty and main contribution of our study (lines 58–63, 427–429, 498–501, 518–524). In the manuscript we cited several articles assessing the SST and chl-a concentration response to the passage of consecutive typhoons in the western Pacific Ocean (e.g., Wu and Li, 2018; Ning et al., 2019; Wang et al 2020). However, to the best of our knowledge there are no previous studies assessing the biological response to consecutive TCs in the western Atlantic Ocean (lines 58–63). In general, the main finding of this work is the additional oceanic variability (i.e., increased ocean cooling and chl-a concentration, deepening of the mixed layer and enhanced upwelling) observed after the passage of a second TC in the western Sargasso Sea indicating that consecutive TCs within a short period of time superimpose effects on the upper ocean response (498–501). In addition, our climatological analysis revealed that the strongest oceanographic response in the western Sargasso Sea is associated with consecutive TCs and long-lasting TC forcing (427–429).

On the other hand, new analyses were performed to make clearer the contribution of consecutive hurricanes superimposing effects on the ocean in the study area. We assessed the individual response induced by Humberto (lines 436–455) and we compared the oceanic response induced by Dorian and Humberto with the one induced by an individual hurricane (Irene, 2011) and other consecutive hurricanes (Dennis and Floyd, 1999) in the study area (lines 456–476). We included a new figure (Figure 13) in the manuscript showing the main results derived from the last analysis mentioned above. Overall, we found that maximum sea surface cooling and chl-a concentration increase induced by Irene occurred immediately after its passage and then cooling and phytoplankton bloom started to decrease from this day onward (lines 468–469). In contrast, when analysing the oceanic response to consecutive Hurricanes Dennis and Floyd, we observe that the most intense cooling and phytoplankton bloom occurred at the end of the third post-storm week (i.e., after the passage of the second TC Floyd) (Figure 13D and F), which is consistent with the temporal evolution of SST and chl-a concentration anomalies retrieved for Dorian and Humberto (lines 473–476).

As we mentioned in the revised version of the manuscript, consecutive TCs superimpose effects on the upper ocean response because of strong-induced mixing and upwelling (lines 476–480). We also mentioned that the maximum deepening of the MLD at the beginning of the third post-storm week (after the passage of Humberto) is consistent with the suggestion made by Ezer et al. (2017), stating

that several storms affecting the same region within a relatively short period of time have a cumulative impact on ocean mixing (lines 354–356).

[Figure]

**Figure 13.** Trajectories of Hurricanes (A) Irene and Katia (2011) and (B) Dennis and Floyd (1999), with colour coding as defined in Figure 1A. Numbers along trajectories indicate the day/month. Daily mean evolution of anomalies of (C and D) sea surface temperature (SST) and (E and F) chlorophyll-a (chl-a) concentration induced by Irene (C and E) and Dennis+Floyd (D and F). The grey shaded area depicts the first and fourth post-storm weeks.

**Minor comments:**

**Lines 203-204: It's unclear what 'respectively' refers to because you mention deepening of 10 and 11 m, along trajectory and in the square area, and days 1 and 2. Is it deepening of 10 and 11 m along the trajectory and in the square area, respectively? Or at day 1 and 2, respectively?**

We have rewritten this sentence as follow: The mixed layer started to deepen 3 days before Dorian's arrival in both study areas, reaching a maximum mean deepening of 10 m along Dorian's trajectory at day 1 and 11 m in the square study area at day 2 (lines 233–235).

**Lines 220-225: It looks like there's a large positive SST trend in Fig. 6A. What if you remove the trend before calculating the 2019 percentages? Do they change?**

In the revised version of the manuscript we included a new analysis in order to remove the positive trend in the time series of SST anomalies shown in Figure 6A (lines 260–267). Removing the positive trend allowed us to compare anomalies in 2019 with the ones in the previous years on the same basis, i.e., without considering the effect of ocean warming in the region. From the detrended time series (Figure 6B), we observe that SST anomalies in 2019 were the coldest ones in the climatological record

except for the ones in 1984 and 1999 confirming the considerable cooling induced by the combined effects of Dorian and Humberto.

[Figure]

**Figure 6.** (A) Time series of weekly mean sea surface temperature (SST) anomalies during the third and fourth post-storm weeks of Dorian in the square study area shown in Figure 1B. (B) Detrended time series of SST anomalies. The dashed lines mark anomalies in 2019 for comparison with the previous years. (C) and (D) Spatially explicit SST anomalies (Dorian+Humberto (2019) induced effects − Climatology (1982–2018)) in the third and fourth post-storm weeks of Dorian, respectively.

**Line 375: Several papers dispute the result of Kossin (2018). These should also be cited (Lanzante, Uncertainties in tropical-cyclone translation speed. Nature., 570, E6–E15, 2019; Yamaguchi et al., Nature Communications, 11, 47, 2020; Moon et al., Climate change and tropical cyclone trend, Nature, 570, E3–E5, 2019)**

We included the suggested references in the revised version of the manuscript (line 426).

**References**

Ezer, T., Atkinson, L. P., and Tuleya, R.: Observations and operational model simulations reveal the impact of Hurricane Matthew (2016) onthe Gulf Stream and coastal sea level, Dynamics of Atmospheres and Oceans, 80, 124–138, 2017.

Ning, J., Xu, Q., Feng, T., Zhang, H., and Wang, T.: Upper ocean response to two sequential tropical cyclones over the northwestern Pacific Ocean, Remote Sensing, 11, 2431, doi:10.3390/rs11202431, 2019.

Wang, T., Zhang, S., Chen, F., MA, Y., Jiang, C., and Yu, J.: Influence of sequential tropical cyclones on phytoplankton blooms in the northwestern South China Sea, Chinese Journal of Oceanology and Limnology, doi:10.1007/s00343-020-9266-7, 2020.

Wu, R. and Li, C.: Upper ocean response to the passage of two sequential typhoons, Deep Sea Research Part I: Oceanographic Research Papers, 132, 68–79, 2018.

---

## Author Comment (AC2) · 11 Dec 2020

The reviewer is thanked for his/her positive and constructive comments on the manuscript.

**REVIEWER 2**

**The study describes the response of the ocean to two consecutive hurricanes in 2019, looking at various aspects, including temperature change, heat content and chlorophyll. The paper is well written and interesting, though the results are not unexpected or especially new. It should be noted though that studies of consecutive hurricanes in this region are not new - the authors may have not been aware for example of some recent studies (that should be cited) of the impact of consecutive hurricanes on the same region, including Todd et al. (2018) on the 2017's hurricanes and Ezer (2020) who studied the same Dorian and Humberto hurricanes of this study! However, the aforementioned papers focused mostly on the impact of hurricanes on coastal sea level and ocean currents (and the Gulf Stream in particular), while here the additional analysis of biological impacts and chlorophyll is an interesting new kink that worth publication.**

We agree with the fact that studies on consecutive hurricanes in this region are not new since, for instance, the articles suggested by this reviewer (cited in lines 39, 44, 320, 324, 327, 465, 480 in the revised version of the manuscript) address the impact of consecutive hurricanes on ocean circulation in the western Sargasso Sea. However, as we mentioned in the manuscript, studies on the effects induced by consecutive TCs have been much less documented as compared to the response to individual TCs (Wu and Li, 2018; Ning et al., 2019) (lines 58–63), mainly because the occurrence of hurricanes following similar trajectories within a short period of time is uncommon. In the manuscript we cited several articles assessing the SST and chl-a concentration response to the passage of consecutive typhoons in the western Pacific Ocean (e.g., Wu and Li, 2018; Ning et al., 2019; Wang et al 2020). However, to the best of our knowledge there are no previous studies assessing the biological response to consecutive TCs in the western Atlantic Ocean (lines 60–63).

**With regards to ocean dynamics, it would be useful to add a little more discussion - the study area is strongly influenced by the Gulf Stream which was significantly weakened during Dorian (Ezer, 2020) and this had implications for mixing and advection, since a weakened GS could also contribute to relative cooling downstream when advection of warm tropical waters slows down.**

**7. Lines 305-312- When discussing the dynamics and mixing processes, may be add something about the role of the Gulf Stream, as in Ezer (2020).**

In the revised version of the manuscript we included the information below in order to discuss the influence of the Gulf Stream on the observed post-storm cooling and mixing (lines 320–329).

The extensive surface cooling observed to the right of Dorian's trajectory (Figure 2B and C) was also reported by Ezer (2020). Besides, this extensive cooling was consistent with the one observed after the passage of Hurricane Matthew (2016) (who followed a similar trajectory as Dorian) and results from the interaction between a hurricane and the core of the Gulf Stream flow and its associated eddy field (Ezer, 2018). Dorian induced a disruption of the Gulf Stream flow, disconnecting the upstream Florida Current from the downstream Gulf Stream and weakening the Gulf Stream flow by almost 50% (Ezer, 2020). This direct effect on surface currents was followed by intense surface cooling largely determined by the reduced flow of warm tropical waters being advected downstream Gulf Stream as well as mixing of the upper oceanic layer (like during the Hurricane Matthew) (Ezer et al., 2017; Ezer, 2020). Overall, the interaction of hurricanes with the Gulf Stream induces a considerable vertical mixing and reduction of the stratification frequency (Kourafalou et al., 2016). Consequently, it has

been reported that vertical mixing drove cooling of the upper 50 m after the passage of Matthew across the Sargasso Sea (Ezer, 2018).

**It would have been interesting also to extend the study period to see the timescale of recovery after the storm and the role that the GS plays in this recovery.**

We assessed the oceanic response during a month after the passage of Dorian in agreement with the methodology followed in previous studies assessing climatological responses to the passage of hurricanes (line 123), which allowed us to compare the retrieved temporal evolution of SST and chl-a concentration anomalies with the ones reported in Menkes et al. (2016) for all TC-prone regions around the world (e.g., lines 431–432, 434–435, 468–470). On the other hand, the analysis of four post-storm weeks allowed us to compare the individual response induced by Dorian in the first and second post-storm weeks with the one induced by the combined effects of Dorian and Humberto in the third and fourth post-storm weeks since the main purpose of our study is to highlight the increased oceanic response to the passage of consecutive TCs. We agree that it would be interesting to analyse the timescale of recovery after the storm and the role that the Gulf Stream plays in this recovery. However, this is out the scope of our research. We consider that a thorough study is needed in order to accurately assess the timescale of the decay of the oceanic response induced by these TCs.

Extending the study period can add uncertainties related with the real environmental drivers governing the retrieved oceanic variability. Oceanic variability within a few days (and weeks) after the passage a hurricane can certainly be associated with the perturbations induced by this atmospheric phenomenon. However, extending the study period increases the probability to account for additional drivers influencing the observed oceanic variability, such as those governing the actual seasonal variability of the analysed variables. It has been reported that TCs occurring in the first half of the hurricane season (at global scale) disrupt the seasonal warming trend, which is not resumed only 20–30 days after the cyclone passage, while cyclone occurrences in the last half of the season lead to an SST decrease from which the ocean does not recover due to the seasonal cooling cycle (Dare and McBride, 2011). On the other hand, Haakman et al. (2019) reported on the basis of climatological analysis that strong cold anomalies induced by a hurricane in the North Atlantic basin need around 50–60 days to (almost) disappear and recover the climatological mean values.

We computed anomalies of SST and chl-a concentration two months after the passage of Dorian (Figure R1). The additional almost 30 days essentially account for October 2019. During this period, SST kept a decreasing trend showing the most negative anomalies of the entire study period. In agreement, chl-a concentration anomalies in the square study area at this time showed the most positive values of the entire analysed post-storm period. We consider that in order to correlate the oceanic variability in October 2019 with the probable long-lasting effects induced by Dorian and Humberto, a climatological analysis would be needed in order to discard the influence of the seasonal cycle of these variables in the study area. This new analysis would extend our study too much. We consider that, definitively, further studies are needed in order to accurately assess the recovery timescale after the passage of consecutives hurricanes in the western Sargasso Sea.

[Figure]

**Figure R1.** Daily mean evolution of anomalies of (A) sea surface temperature (SST) and (B) chlorophyll-a (chl-a) concentration in the Sargasso Sea before, during and after the passage of Dorian and Humberto. The grey-shaded areas depict the extended period of analysis corresponding to October 2019.

**Specific Comments:**

**2. Abstract- Sentence starting "Overall, : : :" (line 7) and ending ": : :post-storm weeks, respectively." (line 12) is too long, cumbersome and needs separating to shorter sentences and rephrasing.**

In the revised version of the manuscript, we have rewritten this sentence as follow: Overall, anomalies of sea surface temperature, ocean heat content and mean temperature from the sea surface to a depth of 100 m were a 50, 63 and 57% smaller (more negative) in the third/fourth post-storm weeks than in the first/second post-storm weeks of Dorian (accounting only for Dorian effects), respectively. For what concerns the biological response, we found that surface chlorophyll-a (chl-a) concentration anomalies, the mean ch-a concentration in the euphotic zone and the chl-a concentration in the deep chlorophyll maximum were 16, 4 and 16% higher in the third/fourth post-storm weeks than in the first/second post-storm weeks, respectively (lines 7–12).

**3. Introduction- The discussion is mostly about climate change and hurricanes, and not enough on the actual topic of the study, i.e., the processes involved in the impact of storms on the upper ocean.**

In the revised version of the manuscript we have included a little more discussion and analyses related with the processes associated with the impact of storms in the upper ocean (e.g., lines 320–329, 436–480). New analyses were performed to make clearer the contribution of consecutive hurricanes superimposing effects on the ocean in the study area as was suggested by the first reviewer. We assessed the individual response induced by Humberto (lines 436–455) and we compared the oceanic response induced by Dorian and Humberto with the one induced by an individual hurricane (Irene 2011) and other consecutive hurricanes (Dennis and Floyd 1999) in the study area (lines 456–476). However, we acknowledge that further studies are needed in order to get a deeper understanding on the processes driving the upper ocean response to the passage of consecutive hurricanes in the study area (lines 516–518).

**4. Line 35- I suggest deleting "In agreement", its awkward to start a paragraph this way.**

In the revised version of the manuscript, we have deleted "In agreement" (line 36).

**5. Lines 38-39- Maybe add citation to Todd et al. (2018) who studied the same mentioned hurricanes, and when discussing Dorian (lines 42, 62, etc.) may cite Ezer (2020).**

We added the suggested citations in several lines in the manuscript (lines 39, 44, 320, 324, 327, 465, 480).

**6. Figures 6a and 11a- It is unclear how they were done and what they represent- how can one compare a hurricane at a particular year and a particular week to past years?, and what does the clear trend represents. It seems that there is much more information in these figures than described and the interpretation is unclear.**

In the revised version of the manuscript, we reformulated the information in lines 248–255 to clarify the procedure followed to create the time series of the weekly anomalies shown in Figures 6A and 11A. Moreover, we have rewritten the results derived from these figures (lines 255–259, 308–311). Overall, we mentioned that the actual third and fourth post-storm weeks of Dorian in the square study area were the ones of 19–26 September 2019 and 27 September–4 October 2019, respectively, so, we assessed the SST and chl-a concentration variability during these weeks in the previous years for which satellite observations were available, i.e., 1982–2018 for SST and 1998–2018 for chl-a concentration. We refer to these weeks as the *third and fourth post-storm weeks of Dorian* regardless of the analysed year, and to indicate the actual oceanic response induced by Dorian and Humberto we specified this was the one in 2019.

On the other hand, the positive trend observed in time series of SST anomalies (Figure 6A in the manuscript) accounts for the ocean surface warming in the region as previously reported (Bulgin et al 2020) (lines 261–262). In the revised version of the manuscript we present a new analysis and results (lines 260–267) since we removed the positive trend in order to compare anomalies in 2019 with the ones in the previous years on the same basis, i.e., without considering the effect of ocean warming in the region. From the detrended time series (Figure 6B), we observe that SST anomalies in 2019 were the coldest ones in the climatological record except for the ones in 1984 and 1999 confirming the considerable cooling induced by the combined effects of Dorian and Humberto.

[Figure]

**Figure 6.** (A) Time series of weekly mean sea surface temperature (SST) anomalies during the third and fourth post-storm weeks of Dorian in the square study area shown in Figure 1B. (B) Detrended time series of SST anomalies. The dashed lines mark anomalies in 2019 for comparison with the previous years. (C) and (D) Spatially

explicit SST anomalies (Dorian+Humberto (2019) induced effects – Climatology (1982–2018)) in the third and fourth post-storm weeks of Dorian, respectively.

**8. Fig. 12c- Are those absolute geostrophic velocity? or velocity anomaly? Anomalies do not tell us much, maybe also show the absolute SSH and velocity to indicate the hurricane track relative to the mean flow of the subtropical gyre and the GS.**

The main purpose of Figure 12 is to depict that in some cases, maximum surface cooling and high chl-a concentration occurred over areas with low values of sea surface height (SSH) and oceanic cyclonic circulation, which largely correspond to cold-core eddies (lines 330–331). In the first version of this manuscript we had used an image of sea surface height anomaly (SSHA) considering that this 'anomaly' preserves seasonal signals since it accounts for the difference between the best estimate of the sea surface height and a mean sea surface derived from long-term observations from satellite altimeters (Muller-Karger et al 2015). For this reason, we can see in Figure R2 that both SSHA and SSH images show a similar spatial distribution of low values of the sea level in the central Sargasso Sea basin. Thus, and considering the suggestion made by this reviewer, we used the image of absolute SSH in the revised version of the manuscript (Figure 12C in the manuscript). We also specified that the vectors of marine currents displayed in Figure 12C represent absolute geostrophic velocity (Figure 12 caption).

[Figure]

**Figure R2.** (A) Sea surface height anomaly (SSHA) and (B) sea surface height (SSH) with the absolute geostrophic velocity vectors superimposed (data derived from Salto/DUACS gridded multimission altimeter at 0.25 × 0.25 spatial resolution). The trajectory of Dorian and a hurricane symbol indicating its estimated position are superimposed on all imagines.

**References**

Bulgin, C. E., Merchant, C. J., and Ferreira, D.: Tendencies, variability and persistence of sea surface temperature anomalies, Scientific Reports, 10, 7986, doi:10.1038/s41598-020-64785-9, 2020.

Dare, R.A., McBride, J.L.: Sea surface temperature response to tropical cyclones. Monthly Weather Review 139 (12), 3798–3808, 2011.

Ezer, T.: On the interaction between a hurricane, the Gulf Stream and coastal sea level, Ocean Dynamics, 68, 1259–1272, 2018.

Ezer, T.: The long-term and far-reaching impact of hurricane Dorian (2019) on the Gulf Stream and the coast, Journal of Marine Systems, p. 103370, doi:10.1016/j.jmarsys.2020.103370, 2020.

Ezer, T., Atkinson, L. P., and Tuleya, R.: Observations and operational model simulations reveal the impact of Hurricane Matthew (2016) on the Gulf Stream and coastal sea level, Dynamics of Atmospheres and Oceans, 80, 124–138, 2017.

Haakman, K., Sayol, J.-M., van der Boog, C. G., and Katsman, C. A.: Statistical characterization of the observed cold wake induced by North Atlantic hurricanes, Remote Sensing, 11, 2368, doi:10.3390/rs11202368, 2019.

Kourafalou, V. H., Androulidakis, Y. S., Halliwell Jr, G. R., Kang, H., Mehari, M. M., Le Hénaff, M., Atlas, R., and Lumpkin, R.: North Atlantic Ocean OSSE system development: Nature Run evaluation and application to hurricane interaction with the Gulf Stream, Progress in Oceanography, 148, 1–25, 2016.

Menkes, C. E., Lengaigne, M., Lévy, M., Éthé, C., Bopp, L., Aumont, O., Vincent, E., Vialard, J., and Jullien, S.: Global impact of tropical cyclones on primary production, Global Biogeochemical Cycles, 30, 767–786, 2016.

Muller-Karger, F.E., Smith, J.P., Werner, S., Chen, R., Roffer, M., Liu, Y., Muhling, B., Lindo-Atichati, D., Lamkin, J., Cerdeira-Estrada, S., Enfield, D.B.: Natural variability of surface oceanographic conditions in the offshore Gulf of Mexico, Progress in Oceanography, 134, 54–76, 2015.

Ning, J., Xu, Q., Feng, T., Zhang, H., and Wang, T.: Upper ocean response to two sequential tropical cyclones over the northwestern Pacific Ocean, Remote Sensing, 11, 2431, doi:10.3390/rs11202431, 2019.

Wang, T., Zhang, S., Chen, F., MA, Y., Jiang, C., and Yu, J.: Influence of sequential tropical cyclones on phytoplankton blooms in the northwestern South China Sea, Chinese Journal of Oceanology and Limnology, doi:10.1007/s00343-020-9266-7, 2020.

Wu, R. and Li, C.: Upper ocean response to the passage of two sequential typhoons, Deep Sea Research Part I: Oceanographic Research Papers, 132, 68–79, 2018.

---

## Author Comment (AC3) · 11 Dec 2020

The reviewer is thanked for his/her positive and constructive comments on the manuscript.

**REVIEWER 3**

**I agree that the introduction focuses more on climate change and TCs and less on upper ocean response. Please add literature and introductory comments on the expectation of upper ocean changes from the passing of hurricanes. An additional paragraph should suffice. Do not take out what you have - it is well written and supports the ideas. You need only add a little more. This will help the remaining parts of your paper pack more of an important "punch" if we are more aware of why it matters to be looking at Chl-a.**

We have included the information below in Introduction in order to briefly describe the upper ocean changes to the passing of TCs (lines 47–58).

Over oceans, TC-induced wind forcing mixes the surface layer, deepens the mixed layer and uplifts the thermocline leading to a decreased upper ocean temperature and heat potential (Price, 1981; Shay and Elsberry, 1987; Trenberth et al., 2018). Vertical mixing and upwelling also lead to an increased abundance of surface phytoplankton due to entrainment of nutrient-rich waters from the nitracline to the ocean surface and/or entrainment of phytoplankton from the deep chlorophyll maximum (DCM) (Babin et al., 2004; Walker et al., 2005; Gierach and Subrahmanyam, 2008; Shropshire et al., 2016). The nutrient influx stimulates phytoplankton growth and can lead to phytoplankton blooms lasting several days after the TC passage in the oligotrophic oceanic waters (Babin et al., 2004; Hanshaw et al., 2008; Shropshire et al., 2016). Moreover, rainfall associated with these extreme meteorological phenomena modulates surface cooling and phytoplankton blooms since rainfall freshens the near-surface water increasing stratification and influencing vertical mixing (Lin and Oey, 2016; Liu et al., 2020). From satellite imagery, an increase in phytoplankton abundance is identified as elevated chlorophyll-a (chl-a) concentration. Distinguishing between mechanisms inducing changes in chl-a concentration is crucial to understanding the impact of storms on upper ocean oceanographic conditions.

**Section 2.1. Please refer to Figure 1 as your "study area graphic", it will lead to your figure being moved up slightly which helps the reader identify the location.**

We mentioned early in the manuscript (line 80, Materials and methods, Subsection 2.1. Study area) that Figure 1 shows our study area.

**Did you do any assessment of other conditions of the atmosphere in the "pre-storm" week of Dorian? Meaning, did it rain at all during that time? Rainfall from minor to severe thunderstorms can alter your SSTs in the region and, without mention of it, I'm not sure that other atmospheric events might be biasing your results.**

The oceanic response induced by a TC is determined by a combination of atmospheric and oceanic variables (Babin et al., 2004). So, we agree with this reviewer that the obtained results are influenced by atmospheric variables that are not directly considered in our study. For instance, as we mentioned in the revised version of the manuscript (lines 53–55), rainfall associated with TCs modulates surface cooling and phytoplankton blooms since rainfall freshens the near-surface water increasing stratification and influencing vertical mixing. However, given the strong impact of TC winds (mentioned in lines 47–48) and rainfall on vertical mixing, entrainment, and upwelling of the thermocline, which in turn modulate the post-storm SST and chl-a concentration response, we consider that through the assessment of the mixed layer depth and the 20 °C isotherm (D20) depth

variability, we indirectly account for the effects of the above-mentioned atmospheric variables. We have included this information in the revised version of the manuscript (lines 190–199).

**You mention TC Erin. Were there any other events? Please comment.**

Thank you for this comment because we realized that, indeed, there was another TC crossing the western Sargasso Sea during our study period. In the revised version of the manuscript we included the trajectory of TC Jerry on Figure 2E. At the end of the third post-storm week and beginning of the fourth one, TC Jerry moved across the central northwestern Atlantic basin as a tropical storm and then weakened to a low-pressure system (Brown, 2019) (Figure 2E). In Figure 2E we can see a patch of considerable low SSTs to the left of Jerry's trajectory (centered at 31 °N and 70 °W approximately), which could have resulted from the combined effects induced by Humberto and Jerry. This patch of low SSTs was located to the right of Humberto's trajectory, who affected this area as a category 3 hurricane (Figure 1A) a week before the passage of Jerry (lines 213–218).

[Figure]

**Figure 2.** Weekly mean sea surface temperature (SST) in the (A) pre-storm week and (B) first, (C) second, (D) third and (E) fourth post-storm weeks of Dorian in the Sargasso Sea. The trajectories of Erin and Jerry are superimposed on (A) and (E), respectively, with colour coding as defined in Figure 1A and arrows indicating their forward movement. The dashed contours in (B) and (E) indicate the probable surface cooling induced by Erin and Jerry, respectively.

**Paragraph including line 190: the way you worded the beginning of this paragraph is confusing. Please rephrase.**

We have rephrased the beginning of the paragraph suggested above (line 219).

**Great Figure 2. Really showcases what you are describing.**

Thank you very much for your positive comment on Figure 2.

**In paragraph with line 275, you bring it up yourself that the findings are similar with Ezer (2018) (or later, Foltz et al. 2015). So why is your study different? And, thus, why is it important that it is published? I think you need a stronger argument than you have presented here. Broader impacts related to your study can be helpful here.**

The extensive surface cooling and the variability of the mixed layer depth found in our study agree with previous reports (e.g., Foltz et al. 2015, Ezer 2018, Ezer 2020) as was mentioned in the manuscript (lines 320–323, 349–350). We consider that the consistency of our results with the ones previously reported helps to confirm and consolidate the knowledge on the oceanic response to TCs in the region, which is needed in order to derive the general behavior of the upper ocean response to TCs. Moreover, the consistency of our results with those previously published confirms the suitability of the dataset used (i.e., a combination of satellite remote sensing and modelled data) to capture the ocean response to the passage of TCs (lines 497–498) since we used a dataset that is different from the ones used in previous studies. On the other hand, in the revised version of the manuscript we emphasized the novelty and main contribution of our study (lines 58–63, 427–429, 498–501, 518–524) on the basis of which we consider it is important to publish it. As we mentioned in Introduction, extensive and long-lasting SST cooling as well as intense post-storm phytoplankton blooms after the passage of consecutive TCs have been documented in the northwestern Pacific Ocean (e.g., Wu and Li, 2018; Ning et al., 2019; Wang et al 2020). However, to the best of our knowledge, there are no previous studies assessing the biological response to consecutive TCs in the western Sargasso Sea (lines 58–63). Insights into the phytoplankton response to severe weather events are essential in order to ascertain the capacity of the oceans to absorb carbon dioxide through photosynthesis (Davis and Yan, 2004) (lines 67–68). Moreover, given that climate-driven processes affecting nutrient availability and phytoplankton primary production affect eel larval survival in areas with high spawning activity such as the Sargasso Sea, we consider that the assessment of the oceanic response to TCs in general, and the biological response, in particular, serves for future studies addressing the influence of climate variability on fishery oceanography in the region (lines 489–494).

**I think your conclusion needs to be strengthened. While you provide a nice summary of what are interesting statistics, you need to relate it to a broader picture. Why does knowing this information help us in some way? Does it inform fisheries? Does it inform management practices? Why does it matter? (I believe it does, but you need to provide a stronger argument for it).**

As we mentioned in Introduction, the assessment of the oceanic response to TCs has been a hot topic given its importance for studies on climate change, ecological variability and environmental protection (lines 65–66). Therefore, in Conclusions we mentioned the importance and application of our results in the research fields mentioned above. Overall, the oceanic response to the passage of Dorian and Humberto reported in our study gives insights into the oceanic implications of a simultaneous increase of both the frequency and intensity of TCs in the North Atlantic basin. Thus, together with a future increase of TC activity in the region, an increased oceanic response could also be expected. Moreover, considering that the TC-induced chl-a concentration changes may impact the survival rates of fish larvae and their recruitment to adulthood in the Sargasso Sea, the results presented here serve for future studies addressing the influence of climate variability on fishery oceanography in the region (lines 519–524).

**References**

Babin, S. M., Carton, J. A., Dickey, T. D., and Wiggert, J. D.: Satellite evidence of hurricane-induced phytoplankton blooms in an oceanic desert, Journal of Geophysical Research: Oceans, 109, C03 043, doi:10.1029/2003JC001938, 2004.

Brown, D. P.: Tropical cyclone report Hurricane Jerry, 17–24 September, 2019, National Hurricane Center, https://www.nhc.noaa.gov/data/tcr, 2019.

Davis, A. and Yan, X.-H.: Hurricane forcing on chlorophyll-a concentration off the northeast coast of the US, Geophysical Research Letters, 31, L17 304, doi:10.1029/2004GL020668, 2004.

Ezer, T.: On the interaction between a hurricane, the Gulf Stream and coastal sea level, Ocean Dynamics, 68, 1259–1272, 2018.

Ezer, T.: The long-term and far-reaching impact of hurricane Dorian (2019) on the Gulf Stream and the coast, Journal of Marine Systems, p.103370, doi:10.1016/j.jmarsys.2020.103370, 2020.

Foltz, G. R., Balaguru, K., and Leung, L. R.: A reassessment of the integrated impact of tropical cyclones on surface chlorophyll in the western subtropical North Atlantic, Geophysical Research Letters, 42, 1158–1164, 2015.

Gierach, M. M. and Subrahmanyam, B.: Biophysical responses of the upper ocean to major Gulf of Mexico hurricanes in 2005, Journal of Geophysical Research: Oceans, 113, C04 029, doi:10.1029/2007JC004419, 2008.

Hanshaw, M. N., Lozier, M. S., and Palter, J. B.: Integrated impact of tropical cyclones on sea surface chlorophyll in the North Atlantic, Geophysical Research Letters, 35, doi:10.1029/2007GL031862, 2008.

Lin, Y.-C. and Oey, L.-Y.: Rainfall-enhanced blooming in typhoon wakes, Scientific Reports, 6, 31310, doi:10.1038/srep31310, 2016.

Liu, F., Zhang, H., Ming, J., Zheng, J., Tian, D., and Chen, D.: Importance of precipitation on the upper ocean salinity response to Typhoon Kalmaegi (2014), Water, 12, 614, doi: 10.3390/w12020614, 2020.

Ning, J., Xu, Q., Feng, T., Zhang, H., and Wang, T.: Upper ocean response to two sequential tropical cyclones over the northwestern Pacific Ocean, Remote Sensing, 11, 2431, doi:10.3390/rs11202431, 2019.

Price, J. F.: Upper ocean response to a hurricane, Journal of Physical Oceanography, 11, 153–175, 1981.

Shay, L. K. and Elsberry, R. L.: Near-inertial ocean current response to Hurricane Frederic, Journal of Physical Oceanography, 17, 1249–1269, 1987.

Shropshire, T., Li, Y., and He, R.: Storm impact on sea surface temperature and chlorophyll a in the Gulf of Mexico and Sargasso Sea based on daily cloud-free satellite data reconstructions, Geophysical Research Letters, 43, 12 199–12 207, 2016.

Trenberth, K. E., Cheng, L., Jacobs, P., Zhang, Y., and Fasullo, J.: Hurricane Harvey links to ocean heat content and climate change adaptation, Earth's Future, 6, 730–744., 2018.

Walker, N. D., Leben, R. R., and Balasubramanian, S.: Hurricane-forced upwelling and chlorophyll a enhancement within cold-core cyclones in the Gulf of Mexico, Geophysical Research Letters, 32, doi:10.1029/2005GL023716, 2005.

Wang, T., Zhang, S., Chen, F., MA, Y., Jiang, C., and Yu, J.: Influence of sequential tropical cyclones on phytoplankton blooms in the northwestern South China Sea, Chinese Journal of Oceanology and Limnology, doi:10.1007/s00343-020-9266-7, 2020.

Wu, R. and Li, C.: Upper ocean response to the passage of two sequential typhoons, Deep Sea Research Part I: Oceanographic Research Papers, 132, 68–79, 2018.